# A Sublinear Adversarial Training Algorithm

**Yeqi Gao**
Tsinghua Univeristy
Beijing, China
`gaoyq23@mails.tsinghua.edu.cn`

**Lianke Qin**
UC Santa Barbara
Santa Barbara, CA, USA
`lianke@ucsb.edu`

**Zhao Song**
Adobe Research
Seattle, WA, USA
`zsong@adobe.com`

**Yitan Wang**
Yale University
New Haven, CT, USA
`yitan.wang@yale.edu`

## Abstract

Adversarial training is a widely used strategy for making neural networks resistant to adversarial perturbations. For a neural network of width $m$, $n$ input training data in $d$ dimension, it takes $\Omega(mnd)$ time cost per training iteration for the forward and backward computation. In this paper we analyze the convergence guarantee of adversarial training procedure on a two-layer neural network with shifted ReLU activation, and shows that only $o(m)$ neurons will be activated for each input data per iteration. Furthermore, we develop an algorithm for adversarial training with time cost $o(mnd)$ per iteration by applying half-space reporting data structure.

## 1 Introduction

After modest adversarial input perturbations, gradient-trained deep neural networks have a tendency to change their prediction results in an incorrect manner (Szegedy et al., 2013). Numerous steps have been taken to build deep neural networks immune to such malicious input. Among such efforts, adversarial training with a min-max object is most effective (Madry et al., 2018) to obtain a robust neural network against perturbed input according to Carlini & Wagner (2017); Athalye et al. (2018). Adversarial training usually yields small robust training loss in practice. The min-max formulation can be viewed as a two-player game. One player is the learner of neural network and the other player is an adversary allowed to arbitrarily perturb the input up to some norm constraint. For every round, the adversary creates adversarial inputs against the existing neural network. Then, the learner adjusts the parameters of neural network by taking a gradient descent step to reduce its prediction loss evaluated by adversarial inputs.

In the past few years, convergence analysis for training neural network on original input has been established. The seminal work of Jacot et al. (2018) initiates the study of *neural tangent kernel* (NTK), which is a very useful analytical model in the deep learning theory area. By over-parameterizing the neural network so that the network width is relatively large $(m \geq \Omega(\text{poly}(n)))$, one can show that the training dynamic on a neural network is almost the same as that on an NTK. Analyzing the convergence guarantee via over-parameterization has been broadly studied (Li & Liang, 2018; Jacot et al., 2018; Du et al., 2019b; Allen-Zhu et al., 2019b;c; Du et al., 2019a; Song & Yang, 2019; Zou et al., 2020; Oymak & Soltanolkotabi, 2020; Lee et al., 2020; Brand et al., 2021; Song et al., 2021a; Huang et al., 2021; Song et al., 2021b; Chen & Xu, 2021; Munteanu et al., 2022; Hu et al., 2022; Zhang, 2022). Inspired by the over-parameterized neural network convergence theory, (Zhang et al., 2020) applies a similar analysis to adversarial training, i.e., training with perturbed input.

Training such adversarial neural networks is usually done via gradient descent, whose time is determined by the product of a number of training iterations and time cost spent on every training iteration. Many previous papers focus on accelerating the time cost per training iteration via nearest neighbor search (Chen et al., 2020; 2021; Daghaghi et al., 2021). For example, SLIDE Chen et al. (2020) speeds up the forward computation by efficiently retrieving activated neurons with the maximum inner product via a locality-sensitive hashing data structure. Using data structure to speed

up optimization algorithms has made much progress for problems like linear programming (Cohen et al., 2021; Lee et al., 2019; Ye, 2020; Song & Yu, 2021; Dong et al., 2021; Jiang et al., 2021), semi-definite programming (Jiang et al., 2020; Huang et al., 2022), sum-of-squares (Jiang et al., 2022), non-convex optimization (Song et al., 2021a; Brand et al., 2021; Song et al., 2021b), reinforcement learning (Shrivastava et al., 2021a), frank-wolfe method (Shrivastava et al., 2021b; Song et al., 2022a) and discrepancy (Song et al., 2022b; Zhang, 2022). Following this line, we will also strive to reduce the time required for each training iteration by designing an algorithm to identify the activated neurons during each training iteration efficiently via high dimensional search data structure.

In this paper, we study a two-layer fully connected neural networks with shifted ReLU $\sigma_\tau : \mathbb{R} \to \mathbb{R}$ activation. We first define the two-layer ReLU activation neural network

$$f(x) := \sum_{r=1}^{m} a_r \cdot \sigma_\tau(\langle w_r, x \rangle + b_r)$$

where $m$ is the number of hidden neurons and $\sigma_\tau(x) := \mathbf{1}[x > \tau] \cdot x$ is the shifted ReLU activation function with threshold $\tau$. $\{w_r\}_{r=1}^{m} \subset \mathbb{R}^d$ is the weight vector, $\{a_r\}_{r=1}^{m} \subset \mathbb{R}$ is the output weight vector, and $x \in \mathbb{R}^d$ is the input vector. The total number of inputs is $n$. Therefore, in each training iteration, we need to compute the forward pass for each input vector which requires $m$ vector inner product of $d$ dimension. This implies a $\Omega(mnd)$ cost per training iteration. One question arises:

*Can we design an adversarial training algorithm only requires $o(mnd)$ cost per iteration?*

The answer is affirmative. We outline our contributions as follows:

- We analyze the convergence of adversarial training for a two-layer neural network with shifted ReLU activation, and show that in each iteration only $o(m)$ of neurons are activated for a single input.
- We have designed an adversarial training algorithm for a two-layer fully-connected neural network with shifted ReLU activation. We leverage a half-space reporting data structure of weights to identify sparsely activated neurons, enabling a sublinear training time cost per iteration.

## 1.1 MAIN RESULT

We give a formal statement of our main result in Theorem 1.1.

**Theorem 1.1** (Time complexity). *Given $n$ data points in $\mathbb{R}^d$ and a neural network model defined in Eq.(2), there exists an adversarial training algorithm (Algorithm 1) whose expected time cost per-iteration is $o(mnd)$.*

**Roadmap.** We present a brief overview of our techniques in Section 2. We present notations and preliminary tools in Section 3. We prove the existence of pseudo-network to approximate the target $f^*$ in Appendix B. We provide the convergence analysis in Appendix C. And we provide some key results in Section 4. We will conduct a time complexity analysis of our adversarial algorithm in Section 5. We conclude the contribution of this paper in Section 6.

## 2 TECHNIQUE OVERVIEW

In this section, we briefly present an overview of the techniques used in this paper.

**Technique I: Approximation via pseudo-network.** An important fact about over-parameterized neural network used in many recent papers is that if a highly over-parameterized neural network $f(x; W) = \sum_{r=1}^{m} a_{r,0} \cdot \sigma_\tau(\langle w_r, x \rangle + b_{r,0})$ has weight close to its random initialization, then by the a *pseudo-network* $g(x; W)$, $f(x; W)$ could be approximated, where $g(x; W)$ is defined as

$$g(x; W) = \sum_{r=1}^{m} a_{r,0} \cdot \langle w_r - w_{r,0}, x \rangle \cdot \mathbf{1}[\langle w_{r,0}, x \rangle + b_{r,0} \geq \tau]$$

We show that $\sup_{x \in \mathcal{X}} |f(x; W) - g(x; W)|$ is very small with a probability $\geq 1 - \theta_c$.[1] We decompose $f(x)$ into three components $f(x) = A(x) + B(x) + C(x)$. $A(x)$, $B(x)$ and $C(x)$ are represented as follows:

$$A(x) := \sum_{r=1}^{m} a_{r,0} \langle \Delta w_r, x \rangle \Phi_{x,r}$$

$$B(x) := \sum_{r=1}^{m} a_{r,0} (\langle w_{r,0}, x \rangle + b_{r,0}) \Phi_{x,r}^{(0)}$$

$$C(x) := \sum_{r=1}^{m} a_{r,0} (\langle w_{r,0}, x \rangle + b_{r,0}) (\Phi_{x,r} - \Phi_{x,r}^{(0)})$$

It follows from triangle inequality that $|f(x; W) - g(x; W)| \leq |A(x) - g(x)| + |B(x)| + |C(x)|$. $\Phi_{x,r}$ and $\Phi_{x,r}^{(0)}$ are defined in Definition 3.6. Then we derive upper bounds for the three terms on the right-hand side respectively. With the exponential failure probability, we could construct an $\epsilon$-net of $\mathcal{X}$ where $\epsilon = \frac{1}{\text{poly}(m)}$ and apply union bound on this $\epsilon$-net. Then we prove the stability of $f$ and $g$, where we get the bound of $|f(x; W) - g(x; W)|$ as $O(K^2 \cdot m^{-1/10})$ with Prob. $1 - \theta_{1/5}$. And we also give the perturbation analysis for $f(x; W)$ and $g(x; W)$ by choosing perturbation $\mu$ with $\|\mu\|_2 \leq \frac{1}{m}$ and $\tau = O(\frac{1}{m})$. The upper bound of $|g(x + \mu; W) - g(x)|$ and $|f(x + \mu; W) - f(x; W)|$ can be bounded by $O(K \cdot m^{-1/2})$ and $O(K \cdot m^{-1} + 1/m^{1/5})$ separately with Pro. $\geq 1 - \theta_{1/2}$. With the pseudo-network at hand, we can give a further analysis of the gradient descent of the $f(x; W)$. In this paper, the convergence of our network $f(x; W)$ is established with the analysis above whereby choosing proper learning rate ($\eta = \Theta(\epsilon m^{-1/5})$) and training iterations ($T = \Theta(\epsilon^{-2} K^2)$), with $W^*$ near the initialization, we can attain

$$\mathcal{L}_{\mathcal{A}^*}(f_{W^*}) + \epsilon \geq \frac{1}{T} \sum_{t=1}^{T} \mathcal{L}_{\mathcal{A}}(f_{W_t})$$

And then, we can show that with Prob. $\geq 1 - \theta_{1/5}$ and $\tau = O(\frac{1}{m})$, the existence of $W^*$ near the initialization where

$$\|W_0 - W^*\|_{2,\infty} \leq \frac{K}{m^{3/5}}$$

and a small enough loss where $\mathcal{L}_{\mathcal{A}^*}(f_{W^*}) \leq \epsilon$, is established.

**Technique II: Shifted ReLU activation.** The second technique is based on the observation that, shifted ReLU activation function on two-layer neural network, the number of activated neurons for each training data point is sublinear in the network width $m$. The Shifted ReLU activation $\sigma_\tau$ is defined as

$$\sigma_\tau(x) := \mathbf{1}[x > \tau] \cdot x.$$

By choosing $\tau$ properly here, we can attain $o(m)$ activated neurons at initialization. Therefore, we first prove that at initialization, the number of activated neurons is upper bounded by $Q_0 = 2m \cdot \exp(-(\frac{K^2}{4m^{1/5}})) = O(m^{c_Q})$ with probability $\geq 1 - n \cdot \exp(-\Omega(m \cdot \exp(-(\frac{K^2}{4m^{1/5}}))))$. Then we analyze the size of neuron set which has different signs at each training iteration compared to the neuron weights at initialization. it can be upper bounded by $O(nm^{7/8})$ with probability at most $\exp(-\Omega(m^{1/5}))$. We can use them to derive the upper bound of activated neurons per training iteration via triangle inequality.

For the correctness of our paper, we also give the analysis of the approximation, the convergence, and the perturbation of our network based on the shifted ReLU function we proposed. Base on the Theorem A.3, we make a bridge between $\mathbf{1}[\langle w_r, x \rangle + b_r > \tau]$ with $\tau = O(\frac{1}{m})$ and $\mathbf{1}[\langle w_r, x \rangle + b_r > 0]$, which is used in the prior work. And with this bridge, we can finish the proof of the analysis above. By choosing $\tau = O(\frac{1}{m})$, we show that $|g(x; W) - f(x; W)| \leq O(K^2 \cdot m^{-1/10})$ with Prob. $\geq 1 - \theta_{1/5}$. According to such a statement, the prediction loss can also be compressed to any $\epsilon \in (0, 1)$ with the

---

[1] We define $\theta_c := \exp(-\Omega(m^c))$.

same requirement on $\tau = O(\frac{1}{m})$ as well as the probability $1 - \theta_{1/5}$. For the perturbation analysis, we also give the bound of $|g(x+\mu;W) - g(x)|$ and $|f(x+\mu;W) - f(x;W)|$ with Pro. $\geq 1 - \theta_{1/2}$ and the Shifted ReLU activation we choose.

In total, in this paper by using Shifted ReLU activation, we attain $o(mnd)$ time complexity per training iteration and give the convergence analysis and the perturbation analysis to make sure the existence of such a network $f(x;W^*)$ which leads to a small enough prediction loss.

**Technique III: Half-space reporting.**  We leverage the data structure designed by Agarwal et al. (1992) to implement the half-space range reporting functionality. The half-space range reporting problem requires us to maintain a data structure to store a finite point set $P \subset \mathbb{R}^d$. In addition to storing $P$, half-space range problem also has queries. Each query can be denoted by $(a, b) \in \mathbb{R}^d \times \mathbb{R}$. For the query $(a, b)$, the data structure should report a subset $P_{a,b} \subset P$ where $P_{a,b}$ is defined as

$$P_{a,b} := \{x \in \mathbb{R}^d : x \in P, \operatorname{sgn}(\langle a, x \rangle - b) \geq 0\}$$

One brute force implementation for the half-space range reporting problem is to maintain $P$ in an array and enumerate all points in $P$ to determine which points are contained in $P_{a,b}$. As we hope to do adversarial training with lower time cost, we introduce a tree data structure to organize points in $P$. With the tree data structure, we could report $P_{a,b}$ efficiently. And we assume that the $a_r$ and $b_r$ are not updated in this paper. In this structure, we use function query$(x, (\tau - b_r))$ to find satisfied $w_r$. And we use $P_{x,(\tau-b_r)}$ as follows

$$P_{x,(\tau-b_r)} = \{w_r \in \mathbb{R}^d : w_r \in P, \operatorname{sgn}(\langle w_r, x \rangle - (\tau - b_r)) \geq 0\}$$

We preprocess the network weights to construct a half-space range reporting (HSR) data structure for $w_r$'s in order that we can efficiently identify the set of activated neurons $Q_{t,i}$ for each of the adversarial input $\widetilde{x}_i$. By Agarwal et al. (1992), we can also attain the time complexity of the functions in the data structure for half-space reporting including init, query, update (See details in Section 5), which is presented as follows

- $\mathcal{T}_{\text{init}}(n, d) = O_d(n \log n)$, $\mathcal{T}_{\text{query}}(n, d, k) = O_d(n^{1-1/\lfloor d/2 \rfloor} + k)$, amortized $\mathcal{T}_{\text{update}} = O_d(\log^2(n))$.

- $\mathcal{T}_{d,\epsilon}(n^{\lfloor d/2 \rfloor + \epsilon})$, $\mathcal{T}_{\text{query}}(n, d, k) = O_d(\log(n) + k)$, amortized $\mathcal{T}_{\text{update}} = O_d(n\lfloor d/2 \rfloor - 1)$.

To get the running time of training per iteration, we compute the time spent on querying the active neuron set for adversarial training data points, forward and backward computation with the activated neuron set, and updating the HSR search data structure respectively. Based on the number of activated neurons at the initialization (where the number of activated neurons is proved to be $o(m)$ by us) and during training, by summing all parts we obtain an algorithm whose running time per iteration is $\widetilde{O}(m^{1-\Theta(1/d)}nd)$.

## 3 PRELIMINARIES

### 3.1 NOTATIONS

In this paper, $\|x\|_p$ denotes the $\ell p$ norm and mainly focuses on $p = 1, 2$ or $\infty$ for a vector $x$.

$B^\top \in \mathbb{R}^{n \times k}$ denotes the transpose of a matrix $B \in \mathbb{R}^{k \times n}$. And we use $\|B\|_1$ as the entry-wise $\ell_1$ norm, $\|B\|$ as the spectral norm and $\|B\|_F$ as the Frobenius norm. $B_j$ denotes the $j$-th column of $B$ where $j \in [n]$. $\|B\|_{2,1}$ denotes $\sum_{j=1}^n \|B_j\|_2$. And $\|B\|_{2,\infty}$ denotes $\max_{j \in [n]} \|B_j\|_2$. We use $\mathcal{N}(\mu, \sigma^2)$ to represent a Gaussian distribution where $\sigma$ is a covariance and $\mu$ is a mean. We use $\theta_c := \exp(-\Omega(m^c))$. We define $\|f\|_\infty$ as follows

$$\|f\|_\infty = \sup_{x \in \mathcal{X}} |f(x)|, \tag{1}$$

where $f : \mathcal{X} \to \mathbb{R}$.

## 3.2 NETWORK FUNCTION

We consider a neural network $f$ which is parameterized by $(a, b, W) \in \mathbb{R}^m \times \mathbb{R}^m \times \mathbb{R}^{d \times m}$:

$$f(x) := \sum_{r=1}^{m} a_r \cdot \sigma_\tau(\langle w_r, x \rangle + b_r) \tag{2}$$

Note that this neural network is two-layer, and also called one-hidden layer. The activation function we consider here is shifted ReLU. $\mathcal{F}$ represent the class of the function above. Because $a \in \mathbb{R}^m$ and $b \in \mathbb{R}^m$ remain constant throughout adversarial training and only $W \in \mathbb{R}^{d \times m}$ is updated, we will also use $f(x; W)$ to denote the network. And with $\tau > 0$, $\sigma_\tau : \mathbb{R} \to \mathbb{R}_{\geq 0}$ is defined as follows

$$\sigma_\tau(x) := \mathbf{1}[x > \tau] \cdot x.$$

Let $\mathcal{S} \subseteq \mathbb{R}^d \times \mathbb{R}$ be the training data with element $(x_i, y_i)$ where $i \in [n]$. There are some standard assumptions in our paper regarding the training set. While maintaining generality, we set for every $x_i \in \mathbb{R}^d$ where $i \in [n]$ that $x_{i,d} = 1/2$ and $\|x_i\|_2 = 1$. Therefore, a set $\mathcal{X} := \{x \in \mathbb{R}^d : x_d = 1/2, \|x\|_2 = 1\}$ is defined. We also define $\mathcal{X}_1$ as a maximal $\frac{1}{k^c}$-net of $\mathcal{X}$ where $c \in \mathbb{R}_+$ is a constant. We set $|y_i| \leq 1$ where $i \in [n]$ for simplicity.

**Definition 3.1** (Initialization of $a$, $W$, $b$). *We define the initialization of $a_0 \in \mathbb{R}^m$ whose entries are uniformly sampled from $\{-\frac{1}{m^{1/5}}, +\frac{1}{m^{1/5}}\}$, $W_0 \in \mathbb{R}^{d \times m}$ whose entries are i.i.d. sampled from $\mathcal{N}(0, \frac{1}{m})$, $b_0 \in \mathbb{R}^m$ whose entries are i.i.d. sampled from $\mathcal{N}(0, \frac{1}{m})$.*

## 3.3 ROBUST AND ADVERSARY LOSS

We take into account the following loss function for analysing the neural nets.

**Definition 3.2** (Lipschitz convex regression loss). *If a loss function $\ell : \mathbb{R} \times \mathbb{R} \to \mathbb{R}$ meets the below criteria, it is a regression loss (Convex and Lipschitz): $\ell(x, x) = 0$ for every $x \in \mathbb{R}$, positive, $1-$Lipshcitz and convex in the second argument.*

**Definition 3.3** ($\rho$-Bounded adversary). *With $\mathcal{B}_2(u, \rho) := \{v \in \mathbb{R}^d : \|v - u\|_2 \leq \rho\} \cap \mathcal{X}$. Suppose $\rho > 0$. For adversary $\mathcal{A} : \mathcal{F} \times \mathcal{X} \times \mathbb{R} \to \mathcal{X}$, we claim $\mathcal{A}$ is $\rho$-bounded, if $\mathcal{A}(f, x, y) \in \mathcal{B}_2(x, \rho)$. And then against a loss function $\ell$, $\mathcal{A}^*$ is defined as the **worst-case** $\rho$-bounded adversary as follows:*

$$\mathcal{A}^*(f, x, y) := \operatorname*{argmax}_{\widetilde{x} \in \mathcal{B}_2(x, \rho)} \ell(y, f(\widetilde{x}))$$

*Given a neural net $f$, we represent an adversarial dataset generated by $\mathcal{A}$ with respect to it as*

$$\mathcal{A}(f, S) := \{(\mathcal{A}(f, x_i, y_i), y_i)\}_{i=1}^n.$$

Then we give formal definition of robust loss function.

**Definition 3.4** (Robust loss and training loss). *For a raw data set $S = \{(x_i, y_i)\}_{i=1}^n$ with $n$ training points and a prediction model $f$, the training loss is*

$$\mathcal{L}(S, f) := \frac{1}{n} \sum_{i=1}^{n} \ell(y_i, f(x_i)).$$

*The robust training loss is defined for $\rho$-bounded adversary $\mathcal{A}$ as*

$$\mathcal{L}_{\mathcal{A}}(f) := \mathcal{L}(\mathcal{A}(f, S), f) = \frac{1}{n} \sum_{i=1}^{n} \ell(y_i, f(\mathcal{A}(f, x_i, y_i)))$$

*In addition, the **worst-case** of the above loss is defined similarly as*

$$\mathcal{L}_{\mathcal{A}^*}(f) := \mathcal{L}(\mathcal{A}^*(f, S), f) = \frac{1}{n} \sum_{i=1}^{n} \max_{\widetilde{x_i} \in \mathcal{B}_2(x_i, \rho)} \ell(y_i, f(\widetilde{x}_i))$$

**Definition 3.5** (Chi-square distribution). *If $a_1, \cdots, a_m$ are independent, standard normal random variables, then the sum of their squares, $Q := \sum_{i=1}^m a_i^2$, is distributed according to the chi-squared distribution with $m$ degrees of freedom.*

---

**Algorithm 1** Sublinear adversarial training

---

 1: **procedure** FASTADVERSARIALTRAINING($a, b$)
 2:     Adversary $\mathcal{A}$
 3:     Learning rate $\eta$.
 4:     Training set $S = \{(x_i, y_i)\}_1^n$
 5:     Initialization $a_0, b_0, W_0$.
 6:     Data Structure for Half-space reporting ds
 7:     ds.INIT($w_1, \cdots, w_m$)
 8:     **for** $t$ in [T] **do**
 9:         $S^{(t)} := \emptyset$
10:         **for** $i$ in [n] **do**
11:             $\widetilde{x}_i^{(t)} = \mathcal{A}(f_{W_t}, x_i, y_i)$
12:             $Q_{t,i} \leftarrow$ ds.QUERY($\widetilde{x}_i^{(t)}$)
13:                        $\triangleright Q_{t,i} \subset [m]$, it is a set of indices $j$ such that $j$-th neuron is activated
14:             $S^{(t)} = S^{(t)} \cup (\widetilde{x}_i^{(t)}, y_i)$
15:         **end for**
16:         $Q_t \leftarrow \cup_{i \in [n]} Q_{t,i}$
17:         Forward pass for $\widetilde{x}_i^{(t)}$ only on neurons in $Q_{t,i}$ for $i \in [n]$
18:         Calculate gradient for $\widetilde{x}_i^{(t)}$ only on neurons in $Q_{t,i}$ for $i \in [n]$
19:         Gradient update $W(t+1) = W(t) - \eta \cdot \nabla_W \mathcal{L}(f_{W(t)}, S^{(t)})$ for the neurons in $Q_t$
20:         ds.DELETE($w_r(t)$) for $r \in Q_t$
21:         ds.ADD($w_r(t+1)$) for $r \in Q_t$
22:     **end for**
23:     **return** $\{W(t)\}_{t=1}^T$
24: **end procedure**

---

**Definition 3.6** (Boolean function for activated neurons). *For $r \in [m]$, we define $\Delta w_r$, $\Phi_{x,r}$ and $\Phi_{x,r}^{(0)}$ as follows:*

$$\Delta w_r := w_r - w_{r,0},$$
$$\Phi_{x,r} := \mathbf{1}[\langle w_{r,0} + \Delta w_r, x \rangle + b_{r,0} \geq \tau],$$
$$\Phi_{x,r}^{(0)} := \mathbf{1}[\langle w_{r,0}, x \rangle + b_{r,0} \geq \tau]$$

### 3.4 WELL-SEPARATED TRAINING SETS

A commonly made assumption in the literature on over-parameterized neural network is to assume the training set is well-separated. There are several variants of separability defined in related literature. We give the formal definition of the $\gamma$-separability adopted in this paper.

**Definition 3.7** ($\gamma$-separability). *If for all $i \neq j \in [n]$, $\gamma \leq \epsilon(\epsilon - 2\rho)$ and $\|x_i - x_j\|_2 \geq \epsilon$, a training dataset $\mathcal{X}$ will be $\gamma$-separable for a $\rho$-bounded adversary.*

### 3.5 SUBLINEAR ADVERSARIAL TRAINING ALGORITHM

Associated with an adversary $\mathcal{A}$, the algorithm 1 can be used to describe training a network in an adversarial way. The adversary creates adversarial samples against the present neural network in the inner loop. To reduce its loss on the new adversarial samples, the neural network's parameter undergoes a gradient descent step in the outer loop.

**Remark 3.8.** *We assume that $S^{(t)}$ and $W_t$ are independent, when calculating the $\nabla_W \mathcal{L}(S^{(t)}, f_{W_t})$ so that we don't need differentiating over $\mathcal{A}$.*

### 3.6 ACTIVATED NEURON SET

In the analysis of our algorithm, we pay attention to the activated neurons. We give the formal definition of activated neuron set in Definition 3.9.

**Definition 3.9** (Activated neuron set). *For all $t \in \{0, 1, \cdots, T\}$, $\tau > 0$ and $i \in [n]$, we define the set which has activated neurons at time $t$ as $Q_{t,i} := \{r \in [m] : \langle w_{r,t}, x_i \rangle + b_r > \tau\}$ The number of activated neurons at time $t$ is also defined as the size of $Q_{t,i}$*

$$k_{i,t} := |Q_{t,i}| \tag{3}$$

## 4 ANALYSIS OF OUR ALGORITHM

In this section, we will provide an analysis from two perspectives. Firstly, we will delve into the convergence analysis when employing the Shifted ReLU activation function. Secondly, we will establish the upper bound of activated neurons during each iteration, a crucial step in our algorithm's time complexity analysis. Now, let's shift our focus to the convergence analysis.

### 4.1 CONVERGENCE ANALYSIS OF ADVERSARIAL TRAINING WITH SHIFTED RELU ACTIVATION

In this section, we provide essential components of the correctness proof. Due to space constraints, we have included the complete proof in the appendix. Here, we offer an overview of our approach to demonstrating the convergence of the training process, considering both adversarial data and the shifted ReLU activation function.

For analyzing the gradient descent of $f(x; W)$, a fake-network $g(x; W)$ is defined in this paper. And we show that by choosing $W$ near the initialization and $\tau = O(\frac{1}{m})$, with Prob. $\geq 1 - \theta_{1/5}$ the neuron network $f(x; W)$ can be approximated by $g(x; W)$ properly. The proof is given in Appendix C.1.

**Theorem 4.1** (Approximation of $f(x; W)$). *With $K \geq 1$, $\tau = O(\frac{1}{m})$, for every $m \geq \text{poly}(d)$, and every $W \in \mathbb{R}^{d \times m}$ where $\|W_0 - W\|_{2,\infty} \leq \frac{K}{m^{3/5}}$ , with probability at most $1/\exp(\Omega(m^{1/5}))$ over the initialization(See Definition 3.1), we have that*

$$\sup_{x \in \mathcal{X}} |g(x; W) - f(x; W)| \geq O(K^2 \cdot m^{-1/10})$$

We do perturbation analysis for $f(x; W)$ and $g(x; W)$ with shifted ReLU activation in the following lemma when the perturbation $\mu$ satisfies that $\|\mu\|_2 \leq \frac{1}{m}$ and $x + \mu \in \mathcal{X}$. We obtain the perturbation upper bound for $|f(x + \mu; W) - f(x; W)|$ and $|g(x + \mu; W) - g(x; W)|$ with the shifted ReLU activation threshold set as $\tau = O(1/m)$, which is proved in Appendix C.9.

**Lemma 4.2** (Perturbation analysis of $f(x; W)$ and $g(x; W)$). *For every $x \in \mathcal{X}_1$, every $\mu \in \mathbb{R}^d$ where $\|\mu\|_2 \leq \frac{1}{m}$ and $x + \mu \in \mathcal{X}$, with $\tau = O(\frac{1}{m})$ and probability at least $1 - 1/\exp(\Omega(m^{1/2}))$, we have*

$$|g(x + \mu; W) - g(x; W)| \leq O(K \cdot m^{-1/2}). \tag{4}$$

*and*

$$|f(x + \mu; W) - f(x; W)| \leq O(K \cdot m^{-1} + 1/m^{1/5}) \tag{5}$$

The convergence of our neural network $f(x; W)$ with shifted ReLU is stated as follows:

**Theorem 4.3** (Optimal weights versus the initialization). *For all $K > 0$, $\epsilon \in (0, 1)$, and for every $m$ larger than $\text{poly}(n, K, 1/\epsilon)$, we set $\eta = \Theta(\epsilon m^{-1/5})$ and $T = \Theta(\epsilon^{-2} K^2)$. in Algorithm 1. For every $W^* \in \mathbb{R}^{d \times m}$ with $\|W_0 - W^*\|_{2,\infty} \leq \frac{K}{m^{3/5}}$, Algorithm 1 outputs weights $\{W_t\}_{t=1}^T$ such that*

$$\mathcal{L}_{\mathcal{A}^*}(f_{W^*}) + \epsilon \geq \frac{1}{T} \sum_{t=1}^T \mathcal{L}_{\mathcal{A}}(f_{W_t})$$

*with succeed probability at least $1 - \theta_{1/5}$. The randomness is from random initialization (See Definition 3.1).*

In the following statement, we show that by choosing proper $\tau = O(\frac{1}{m})$ and $m$ as a large enough constant, a $W^*$ around the initialization and a small enough worst-case robust loss can be attained by us. The proof is given in Appendix B.8.

**Theorem 4.4** (Convergence of Algorithm 1). *For all $\epsilon \in (0, 1)$, we can get $K = \text{poly}((n/\epsilon)^{1/\gamma})$ and let $m$ be larger than some $\text{poly}(d, (n/\epsilon)^{1/\gamma})$ we can find, with probability $\geq 1 - \theta_{1/5}$ and $\tau = O(\frac{1}{m})$, there will be a $W^* \in \mathbb{R}^{d \times m}$ such that $\mathcal{L}_{\mathcal{A}^*}(f_{W^*}) \leq \epsilon$ and $\|W_0 - W^*\|_{2,\infty} \leq \frac{K}{m^{3/5}}$. The randomness is because of the selection of $W_0, b_0, a_0$.*

## 4.2 THE UPPER BOUND OF ACTIVE NEURONS DURING TRAINING

For the purpose of getting the time complexity of per training iteration, we propose the following two statements, Lemma 4.5 and Claim 4.6. And their proofs are given in Appendix C.13 and C.12. By applying shifted ReLU activation $\sigma_\tau$ in neural network and choosing $K/m^{3/5}$ as $\tau$, we can attain $o(m)$ activated neurons here. With $o(m)$ activated neurons, we can attain the running time per training iteration.

**Lemma 4.5** (Number of activated neurons during initialization). *Let $c_Q \in (0,1)$ denote a fixed constant. Let $Q_0 = 2m \cdot \exp(-(\frac{K^2}{4m^{1/5}})) = O(m^{c_Q})$. For every $\eta \in \mathbb{R}_+$, $x \in \mathcal{X}$ and $r \in [m]$, $\Psi_r(x,\eta)$ is defined as $\Psi_r(x,\eta) := \mathbf{1}[|\langle w_{r,0}, x \rangle + b_{r,0}| \geq \eta]$. With succeed probability at least $1 - n/\exp(\Omega(m \cdot \exp(-(\frac{K^2}{4m^{1/5}}))))$, it holds that*

$$\sum_{r \in [m]} \Psi_r(x, K/m^{3/5}) \leq Q_0.$$

Besides the activated neurons during initialization bounded, we also give the bound of activated neurons versus the initialization. The proof given in Appendix C.12 is based on the shifted ReLU activation $\sigma_\tau$ with $\tau > 0$. The sublinear activation of neurons, along with the limitation on the number of neurons versus the initialization which is the primary focus of our computations during the training, provides us with an opportunity to employ the Half-Space Data Structure introduced in the following section. This allows us to maintain a time complexity of $o(nmd)$.

**Claim 4.6** (Bound for activated neurons versus the initialization). *For any $\|\Delta w_r\|_2 \leq m^{-15/24}$ and any subset $\{x_i\}_1^n \subseteq \mathcal{X}$. Then it holds with succeed probability at least $1 - \theta_{1/5}$ that*

$$\sum_{r=1}^m \mathbf{1}[\exists i \in [n], \ \mathrm{sgn}(\langle x_i, w_{r,0} + \Delta w_r \rangle + b_{r,0} - \tau) \neq \mathrm{sgn}(\langle x_i, w_{r,0} \rangle + b_{r,0} - \tau)] < O(nm^{7/8})$$

$$\forall i \in [m], \mathbf{1}[\exists i \in [n], \ \mathrm{sgn}(\langle x_i, w_{r,0} + \Delta w_r \rangle + b_{r,0} - \tau) \neq \mathrm{sgn}(\langle x_i, w_{r,0} \rangle + b_{r,0} - \tau)] < O(nm^{-1/8})$$

*The randomness is from initialized states (See Definition 3.1).*

## 5 TIME COMPLEXITY

For the identification of activated neurons at each iteration, we will introduce a half-space data structure. This section 5 analyzes the adversarial training algorithm's time complexity. In section 5.1, we provide the Half-space reporting data structure at first. In Section 5.2, we examine the time complexity of our sublinear time adversarial training algorithm.

## 5.1 DATA STRUCTURE FOR HALF-SPACE REPORTING

We formally give the definition of the problem of the half-space range reporting, which is of importance in the field of computational geometry. And we give the data-structure in Agarwal et al. (1992) whose functions are outlined in Algorithm 2 and complexity is given in Corollary 5.2.

**Definition 5.1** (Half-space range reporting). *For a set of $m$ points $P \subseteq \mathbb{R}^d$, three operations are supported:*

- QUERY($W$): *find each point in $P \cap W$ with $W \subset \mathbb{R}^d$ as a half-space.*

- INSERT($p$) : *add a point $p$ to $P$.*

- REMOVE($p$) : *remove a point $p$ from $P$.*

We address the problem (See Definition 5.1) with the data-structure in Agarwal et al. (1992) which has the functions outlined in Algorithm 2. Because the data-structure partitions the set $P$ recursively as well as uses a tree data-structure to organizes the points. Given a query $(b, c)$, all the points $z \in \mathbb{R}^d$ from $P$ which satisfy $\langle b, z \rangle - c \geq 0$ can be quickly found from $P$. And in our paper, the problem is finding all $r \in [m]$ such that $\langle w_r, z \rangle + b_r - \tau > 0$. The Half-Space (See Definition 5.1.) in our algorithm is defined by $(z, (\tau - b_r))$ for $r \in [m]$ and the QUERY($b, c$) holds for QUERY($w_r, (\tau - b_r)$).

---

**Algorithm 2** Data Structure For Half Space Reporting

---
1: **data structure**
2:      INIT$(P, n, d)$               ▷ Construct our data structure via $P \subseteq \mathbb{R}^d$, $|P| = n$
3:      QUERY$(b, c)$    ▷ $b, c \in \mathbb{R}^d$. Find all the points $z \in P$ which satisfies sgn$(\langle b, z \rangle - c) \geq 0$
4:      INSERT$(z)$                               ▷ Insert a point $z \in \mathbb{R}^d$ to $P$
5:      REMOVE$(z)$                            ▷ Remove a point $z \in \mathbb{R}^d$ from $P$
6: **end data structure**

---

**Corollary 5.2** (Agarwal et al. (1992))**.** *Considering $n$ points in $R^d$ as a set, we can solve the half-space reporting problem with the complexity as follows, where $\mathcal{T}_{\mathrm{INIT}}$ indicates the time to construct the data structure, $\mathcal{T}_{\mathrm{QUERY}}$ represent the time spent on each query and $\mathcal{T}_{update}$ represent the time on each update:*

- *$\mathcal{T}_{init}(n, d) = O_d(n \log n)$, $\mathcal{T}_{query}(n, d, k) = O_d(n^{1-1/\lfloor d/2 \rfloor} + k)$, amortized $\mathcal{T}_{update} = O_d(\log^2(n))$.*

- *$\mathcal{T}_{d,\epsilon}(n^{\lfloor d/2 \rfloor + \epsilon})$, $\mathcal{T}_{query}(n, d, k) = O_d(\log(n) + k)$, amortized $\mathcal{T}_{update} = O_d(n \lfloor d/2 \rfloor - 1)$.*

## 5.2 WEIGHTS PREPROCESSING

We offer an adversarial training algorithm with sublinear time in this part. The foundation of our algorithm (Algorithm 1) is the construction of a *search* data structure (in high dimensional space) for neural network weights. To avoid needless calculation, the activated neurons can be identified quickly by using the Half-Space reporting technique. We provide an algorithm with preprocessing procedure for $w_r$ where $r \in [m]$, which is widely used (see Chen et al. (2020); Kitaev et al. (2020); Chen et al. (2021)). The two-layer ReLU network is given in section 3.2 as $f(x) = \sum_{r=1}^{m} a_r \cdot \sigma_\tau(\langle w_r, x \rangle + b_r)$. We can rapidly identify a collection of active neurons $Q_{t,i}$ for every $i$ (which is the adversarial training sample $x_i$) by building a HSR data structure. Pseudo-code is shown in Algorithm 1. The time complexity analysis of the algorithm is the major topic of the remaining portion of this section.

## 5.3 TIME COMPLEXITY PER TRAINING ITERATION

Since we leverage the Half-Space Data Structure, the training time complexity is inherently linked to the activated neurons, as elaborated in Section 4.2. Given that the number of activated neurons is sublinear during each iteration, the following conclusion naturally ensues.

**Lemma 5.3** (Time complexity)**.** *Given a two-layer ReLU network defined in Eq.(2). Assume there are $n$ data points. Say those points are from $d$-dimensional space. Then, the expected time cost for each iteration of our adversarial algorithm (Algorithm 1) runs in time $\widetilde{O}(m^{1-\Theta(1/d)} nd)$.*

The detailed proof is deferred to Appendix C.16.

## 6 CONCLUSION

Making neural networks more resilient to the impact of adversarial perturbations is a common application of adversarial training. The usual time complexity of training a neural network of width $m$, and $n$ input training data in $d$ dimension is $\Omega(mnd)$ per iteration for the forward and backward computation. We show that only a sublinear number of neurons are activated for every input data during every iteration when we apply the shifted ReLU activation layer. To exploit such characteristics, we leverage a high-dimensional search data structure for half-space reporting to design an adversarial training algorithm that only requires $\widetilde{O}(m^{1-\Theta(1/d)} nd)$ time for every training iteration. To the best of our knowledge, our work does not result in adverse effects on society. We also emphasize that the training procedure proposed in this paper is equivalent to applying adversarial training to neural network. Although there is no extra negative societal impact introduced by our method, to make sure the neural network model is properly used, the user needs extra attention, which goes beyond the scope of this paper.

ACKNOWLEDGMENTS

Yitan Wang gratefully acknowledges support from ONR Award N00014-20-1-2335 and a Simons Investigator award from the Simons Foundation to Daniel Spielman.

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

APPENDIX

**Roadmap.** We first present some useful tools from previous work in Section A, then demonstrate the existence of pseudo-network which can approximate $f^*$ in Section B. We then present our convergence analysis in Section C.

## A    MORE TOOLS FROM PREVIOUS WORK

**Lemma A.1** (Corollary 5.4 in Frostig et al. (2016)). *With $k = \frac{1}{\eta^2} \ln(2/\epsilon_1)$ and $x \in [-1, \eta] \cup [\eta, 1]$, we have $|\operatorname{sgn}(x) - p_k(x)| \le \epsilon_1/2$, where $p_k(z) := z \sum_{i=0}^{k} (1 - z^2)^i \prod_{j=1}^{i} \frac{2j-1}{2j}$.*

**Theorem A.2** (Theorem 3.3 from Sachdeva et al. (2014)). *With $X_1, \ldots, X_m$ as independent $\pm 1$ random variables, $X_m := \sum_{i=1}^{m} X_i$ and $t \ge 0$, for all $m, t$ which are positive integers and all $z \in [-1, 1]$, we have*

$$|\mathbb{E}_{X_1, \ldots, X_m}[T_{Z_m}(z) \mathbf{1}[|X_m| \le t]] - z^m| \le 2e^{-t^2/(2m)}$$

**Theorem A.3** (Theorem 3.1 from Li & Shao (2001)). *Suppose that $\eta > 0$ and $\tau > 0$. We obtain,*

$$\exp(-\eta^2/2) \Pr_{x \sim \mathcal{N}(0,1)}[|x| \le \tau] \le \Pr_{x \sim \mathcal{N}(0,1)}[|x - \eta| \le \tau] \le \Pr_{x \sim \mathcal{N}(0,1)}[|x| \le \tau].$$

The following results will be used in our proof.

**Definition A.4** (Lipschitz continuity). *A function $f : X \to Y$ is called $C$-Lipschitz continuous if, for each $x_1, x_2 \in X$ it holds that $d(f(x_1), f(x_2)) \le C \cdot d(x_1, x_2)$, where $d$ stands for the distance.*

**Lemma A.5.** *With $X$ is sampled from $\mathcal{N}(0, \sigma^2)$, we have $\Pr[|X| \le \eta] \in (\frac{2}{3} \frac{\eta}{\sigma}, \frac{4}{5} \frac{\eta}{\sigma})$.*

**Lemma A.6** (Hoeffding bound Hoeffding (1963)). *For $\{X_i\}_1^n$ where $X_i$ is bounded by $[c_i, d_i]$ independently, $\sigma_i = (d_i - c_i)$ and $U = \sum_{i \in [n]}^{n} X_i$, it holds that $\Pr[|U - \mathbb{E}[U]| \ge \eta] \le 2\exp(-\frac{2\eta^2}{\sum_{i \in [n]} \sigma_i^2})$.*

**Lemma A.7** (A Sharper Bound for a Chi-square Variable). *Let $X$ be chi-square with $n$ degrees of freedom. For any positive $t$, the tail of $X$ can be bounded by $\Pr[X - n \ge 2\sqrt{nt} + 2t] \le e^{-t}$ and $\Pr[X - n \le -2\sqrt{nt}] \le e^{-t}$.*

**Lemma A.8** (Chernoff bound Chernoff (1952)). *With $Y = \sum_{i=1}^{n} Y_i$, where $\Pr[Y_i = 1] = p_i$ and $\Pr[Y_i = 0] = 1 - p_i$ for all $i \in [n]$, and $Y_i$ are independent, and $\mu = \mathbb{E}[Y] = \sum_{i=1}^{n} p_i$, it holds that*

- $\forall 0 < \epsilon < 1, \Pr[Y \le \mu(1 - \epsilon)] \le \exp(-\epsilon^2 \mu/2)$,

- $\forall \epsilon > 0, \Pr[Y \ge \mu(1 + \epsilon)] \le \exp(-\epsilon^2 \mu/3)$.

**Lemma A.9** (Bernstein inequality Bernstein (1924)). *We use $Y_1, \cdots, Y_n$ to represent independent random variables whose mean are $0$. Assume that $|Y_i| \le M$ almost certainly. We have for every $\eta > 0$,*

$$\Pr[\sum_{i=1}^{n} Y_i > \eta] \le \exp\left(-\frac{\frac{1}{2}\eta^2}{\sum_{j=1}^{n} \mathbb{E}[Y_j^2] + \frac{1}{3}M\eta}\right).$$

**Claim A.10.** *Let $w, b$ be sampled from $\mathcal{N}(0, I_d)$ and $\mathcal{N}(0, 1)$ respectively. Suppose $\eta \ge 0$ and for every $x \in \mathcal{X}$, it holds that $\Pr[|\langle w, x \rangle + b| \le \eta] = O(\eta)$.*

## B    EXISTENCE OF PSEUDO-NETWORK TO APPROXIMATE $f^*$

In this section, a complexity measure's definition for polynomials is given first in Section B.1. In Section B.2, we state a tool from previous work. We demonstrate the existence of a function $f^*$ that reliably fits the training dataset and has "low complexity" in Section B.3. We prove that there exists a univariate polynomial $q_{\epsilon_1}(z)$ to estimate the step function in Section B.4. We provide a stronger version of the polynomial approximation guarantee in Section B.5. An upper bound of the coefficients of $C_k(x)$ is given in Section B.6.

Then we demonstrate that by using a pseudo-network we can approximate $f^*$ in Section B.7. We put them together to prove Theorem B.10 in Section B.8.

## B.1 DEFINITIONS

In the beginning, we define a complexity measure of polynomials according to Allen-Zhu et al. (2019a) with $K = 1$ in this work.

**Definition B.1.** *With parameter $\epsilon_1 > 0$, $c$ as a large enough constant and univariate polynomial $\phi(z) = \sum_{j=0}^{k} \alpha_j z^j$ with any degree-$k$, the following two complexity measures are what we define*

$$\mathcal{C}(\phi, \epsilon_1) := \sum_{j=0}^{k} c^j \cdot (1 + (\sqrt{\ln(1/\epsilon_1)/j})^j) \cdot |\alpha_j|$$

*and*

$$\mathcal{C}(\phi) := c \cdot \sum_{j=0}^{k} (j+1)^{1.75} |\alpha_j|.$$

## B.2 TOOLS FROM PREVIOUS WORK

We state a tool from previous work (Laurent & Massart, 2000).

**Lemma B.2** (Laurent-Massart (Laurent & Massart, 2000))**.** *Let $a_1, ..., a_n$ be nonnegative, and set*

$$|a|_{\infty} = \sup_{i \in [n]} |a_i|, |a|_2^2 = \sum_{i=1}^{n} a_i^2$$

*For i.i.d $Z_i \sim N(0,1)$, let $X = \sum_{i=1}^{n} a_i(Z_i^2 - 1)$. Then the following inequalities hold for any positive $t$:*

$$\Pr[X \geq 2|a|_2^2 \sqrt{t} + 2|a|_{\infty}] \leq e^{-t}$$
$$\Pr[X \leq -2|a|_2 \sqrt{t}] \leq e^{-t}$$

**Lemma B.3** (Lemma 6.2 from Allen-Zhu et al. (2019a))**.** *Let $\mathcal{C}$ be defined as Definition B.1. For every $\epsilon_2 \in (0, 1/\mathcal{C}(\phi))$. Let $\phi : \mathbb{R} \to \mathbb{R}$ be a univariate polynomial. There will be a function $h : \mathbb{R}^2 \to [-\mathcal{C}(\phi, \epsilon_2), \mathcal{C}(\phi, \epsilon_2)]$ so that for all $y, w^* \in \mathbb{R}^d$ where $\|y\|_2 = \|w^*\|_2 = 1$: the following*

$$\left| \mathbb{E}[h(\langle w^*, u \rangle, \beta) \cdot \mathbf{1}\{\langle u, y \rangle + \beta \geq 0\}] - \phi(\langle w^*, y \rangle) \right| \leq \epsilon_2.$$

*holds. In the above equation, $\beta \sim \mathcal{N}(0,1), u \sim \mathcal{N}(0, I_d)$.*

## B.3 ROBUST FITTING WITH POLYNOMIALS

The purpose of this section is to give the proof of Lemma B.4.

**Lemma B.4** (Lemma 6.2 in Zhang et al. (2020))**.** *With $M = 24\gamma^{-1} \ln(48n/\epsilon)$, we have a polynomial $q : \mathbb{R} \to \mathbb{R}$ where the upper bound of its coefficients is $O(\gamma^{-1} 2^{6M})$ and the upper bound of its degree is $M$, such that for every $\widetilde{x}_j \in \mathcal{B}_2(x_j, \rho)$ and $j \in [n]$,*

$$\left| \sum_{i=1}^{n} y_i \cdot q(\langle x_i, \widetilde{x}_j \rangle) - y_i \right| \leq \epsilon/3.$$

## B.4 EXISTENCE OF STEP FUNCTION ESTIMATION

In the following claim, we prove the existence of a univariate polynomial $q_{\epsilon_1}(z)$ to roughly estimate step function.

**Claim B.5** (Lemma 6.4 in Zhang et al. (2020))**.** *With $M = \frac{24 \ln(16/\epsilon_1)}{\gamma}$ and $\epsilon_1 \in (0, 1)$, there will be a univariate polynomial $q_{\epsilon_1}(z)$ with the bound of coefficients is $O(\frac{2^{6M}}{\gamma})$ and degree at most $M$, such that*

1. *$|q_{\epsilon_1}(z)| \leq \epsilon_1$, for all $z$ such that $-1 \leq z < 1 - (\delta - \rho)^2/2$.*

2. *$|q_{\epsilon_1}(z) - 1| \leq \epsilon_1$, for all $z$ such that $1 - \rho^2/2 \leq z \leq 1$*

### B.5 STRONGER POLYNOMIAL APPROXIMATION

According to Frostig et al. (2016), we need a more robust version of the sgn function's polynomial approximation result in the following lemma.

**Lemma B.6** (Lemma 6.4 in Zhang et al. (2020)). *Let $M = \frac{3}{\eta} \ln(\frac{2}{\eta \epsilon_1})$ and $\eta, \epsilon_1 \in (0, 1)$. There will be a univariate polynomial*

$$p_{\epsilon_1}(x) = \sum_{j \in \{0, 1, \cdots, k\}} \alpha_j x^j$$

*with $|\alpha_j| \le 2^{4M}$ and degree $k \le M$, which is the sgn function's $\epsilon_1$-approximation in $[-1, 1] \setminus (-\eta, \eta)$ such that*

1. *$\forall x \in [-1, -\eta], |p_{\epsilon_1}(x) + 1| \le \epsilon_1$.*

2. *$\forall x \in [\eta, 1], |p_{\epsilon_1}(x) - 1| \le \epsilon_1$.*

### B.6 UPPER BOUND OF SIZE OF COEFFICIENTS OF $C_k(x)$

In the following we get the upper bound of coefficients of $C_k(x)$'s size.

**Proposition B.7** (Proposition A.13 in Zhang et al. (2020)). *We define $C_k(x)$ as follows:*

$$C_k(x) := \sum_{i=0}^{\lfloor 0.5k \rfloor} \binom{k}{2i} \sum_{j \in \{0, \cdots, i\}} \binom{i}{j} x^{2j} (-1)^{i-j} x^{k-2i}$$

*The size of the coefficients of $C_k(x)$ is at most $2^{2k}$.*

The following outcome is a direct result of the Proposition B.7:

**Corollary B.8** (Corollary A.14 in Zhang et al. (2020)). *With $s > 0$, $M_s := \sum_{i=1}^{s} X_i$, $M \ge 0$ and $X_1, \ldots, X_s$ iid $\pm 1$ random variables, $p_{s,M}(x)$ is defined as:*

$$p_{s,M}(x) := \mathbb{E}_{X_1, \ldots, X_s}[C_{M_s}(x) \mathbf{1}[|M_s| \le M]].$$

*The upper bound of $p_{s,M}(x)$'s degree is $M$ and the upper bound of its coefficients is $2^{2M}$.*

### B.7 FAKE-NETWORK APPROXIMATES $f^*$

Suppose $m > \text{poly}(d, (n/\epsilon)^{1/\gamma})$. For a matrix $W^* \in \mathbb{R}^{d \times m}$, with a fake-network $g(x; W^*)$, $f^*$ can be approximated uniformly across $\mathcal{X}$. Compared to Zhang et al. (2020), our neuron network here is based on the shifted ReLU activation whose threshold is $\tau$.

**Lemma B.9** (A variation of Lemma 6.5 in Zhang et al. (2020)). *For all $\epsilon \in (0, 1)$, there will be $K = \text{poly}((n/\epsilon)^{1/\gamma})$ so that let $\tau = O(\frac{1}{m})$, $m$ is larger than some $\text{poly}(d, (n/\epsilon)^{1/\gamma}$ we can find and then with probability $\ge 1 - 1/\exp(\Omega((\frac{m}{n})^{1/2}))$, there will be a $W^* \in \mathbb{R}^{d \times m}$ such that*

$$\sup_{x \in \mathcal{X}} |g(x; W^*) - f^*(x)| \le \epsilon/3$$

*and*

$$\|W^* - W_0\|_{2,\infty} \le \frac{K}{m^{3/5}}$$

*Recall that the randomness is due to initialization (See Definition 3.1).*

*Proof.* With $q(z)$ as the polynomial constructed from Lemma B.4 for every $i \in [n]$, its complexities $\mathcal{C}(q)$ and $\mathcal{C}(q, \epsilon_3)$ (See Definition B.1) can be bounded now, where we will set $\epsilon_3$ after $\mathcal{C}(q)$ bounded ($\epsilon_3 < 1/\mathcal{C}(q)$ according to Lemma C.18).

Next, we explain how to upper bound the $\mathcal{C}(q)$, one can get

$$\mathcal{C}(q) \le c \cdot c_2 \sum_{j=0}^{M} (j+1)^{1.75} \frac{1}{\gamma} 2^{6M}$$

$$< c \cdot c_2 \frac{(M+1)^{2.75}}{\gamma} 2^{6M}.$$

where the 1st inequality is due to Definition B.1 and $M$ is the upper bound of the degree of $q$ and $c_2 \frac{1}{\gamma} 2^{6M}$ is the upper bound of the size of its coefficients, where $c_2 \in \mathbb{R}_+$ is a constant and $M = \frac{24}{\gamma} \ln(48n/\epsilon)$, the second step is due to $(M+1)^{2.75} \geq \sum_{j=0}^{M} (j+1)^{1.75}$.

By $\epsilon_3 = (c \cdot c_2 \frac{(M+1)^{2.75}}{\gamma} 2^{6M})^{-1}$, we can get:

$$\begin{aligned} \ln(1/\epsilon_3) &= \ln(c \cdot c_2 \frac{(M+1)^{2.75}}{\gamma} 2^{6M}) \\ &\leq O(M) \end{aligned} \tag{6}$$

Then we discuss about how to upper bound $\mathcal{C}(q, \epsilon_3)$,

$$\begin{aligned} \mathcal{C}(q, \epsilon_3) &\leq c_2 \sum_{j=0}^{M} \frac{1}{\gamma} 2^{6M} c^j (1 + \sqrt{\ln(1/\epsilon_3)/j})^j \\ &\leq O(1) \frac{1}{\gamma} 2^{6M} (M+1) c^M e^{\sqrt{M \ln 1/\epsilon_3}} \\ &\leq O(1) \frac{1}{\gamma} 2^{6M} (M+1) c^M e^{\sqrt{M \cdot O(M)}} \\ &\leq 2^{O(M)} \end{aligned}$$

where the first step is due to Definition B.1 and $M$ is the upper bound of the degree of $q$ and $c_2 \frac{1}{\gamma} 2^{6M}$ is the upper bound of the size of its coefficients, the second step comes from the geometric sum formula, the third step comes from Eq. (6), and the fourth step follows that $c^M e^{\sqrt{M \cdot O(M)}} \leq 2^{O(M)}$.

With regard to the selection of $k$ and the random variables, we now describe how to carry out the $n$ use of the lemma.

We define

$$\widetilde{B} := \lceil c_1 \frac{d}{\epsilon_3^2} \mathcal{C}^2(q, \epsilon_3) \rceil.$$

We have that $n\widetilde{B} \leq d(\frac{n}{\epsilon})^{c/\gamma} \leq m$ with a large enough $c$ chosen. We use the Lemma C.18 with $k = \lfloor \frac{m}{n} \rfloor$ for $i \in [n-1]$ and with $k = m - (n-1)\lfloor \frac{m}{n} \rfloor$ for $i = n$.

For the $i$-th datapoint, we use the $(m^{1/2}W_{(i-1)\lfloor \frac{m}{n} \rfloor+r,0}, m^{1/2}b_{(i-1)\lfloor \frac{m}{n} \rfloor+r,0}, a_{(i-1)\lfloor \frac{m}{n} \rfloor+r,0})$ as $(w_{r,0}, b_{r,0}, a_0)$ in the Lemma C.18.

By using a union bound over $n$ different terms, we can recompute the probability, which is

$$\geq 1 - n/\exp(\Omega(\sqrt{m/n})) = 1 - 1/\exp(\Omega(\sqrt{m/n})).$$

Recall the movement of weight,

$$\Delta W = [\Delta W^{(1)}, \cdots, \Delta W^{(m)}] \in \mathbb{R}^{d \times m}.$$

Then we can obtain the following upper bound for $\|\Delta W\|_{2,\infty}$:

$$\begin{aligned} \|\Delta W\|_{2,\infty} &\leq O(\frac{\mathcal{C}(q, \epsilon_3) m^{1/5}}{\lfloor \frac{m}{n} \rfloor}) \\ &\leq O(m^{-4/5} \cdot \mathcal{C}(q, \epsilon_3) \, n) \\ &\leq O(m^{-3/5} \cdot \mathcal{C}(q, \epsilon_3) \, n) \\ &\leq O(m^{-3/5} \cdot (n/\epsilon)^{O(\gamma^{-1})}) \end{aligned}$$

where the reason behind the first inequality is Lemma C.18, the second inequality is due to simplifying the terms, the third step is due to $\frac{1}{m^{4/5}} \leq \frac{1}{m^{3/5}}$ and the final step comes from $\tilde{B}n \leq (n/\epsilon)^{c/\gamma}d \leq m$.

And

$$\forall x \in \mathcal{X}, \ |\sum_{r=1}^{m} \mathbf{1}[\langle w_{r,0}, x \rangle + b_{r,0} \geq \tau]a_{r,0}\langle \Delta w_r, x \rangle - \sum_{i=1}^{n} y_i q(\langle x_i, x \rangle)| \leq n\epsilon_3 \leq \epsilon/3,$$

where the final inequality serves as a rough but adequate bound. □

## B.8 PUTTING IT ALL TOGETHER

When we construct function the $g_{x;W^*}$ and $f(x; W)$, we bring the new shifted RELU activated function. And existence problem is base on the new function constructed compared to Zhang et al. (2020). Put everything in this section together we can prove the following theorem:

**Theorem B.10** (A variation of Theorem 5.3 in Zhang et al. (2020)). *For all $\epsilon \in (0, 1)$, we can get $K = \text{poly}((n/\epsilon)^{1/\gamma})$ and let $m$ be larger than some $\text{poly}(d, (n/\epsilon)^{1/\gamma})$ we can find and $\tau = O(\frac{1}{m})$, with probability $\geq 1 - \theta_{1/5}$ , there will be a $W^* \in \mathbb{R}^{d \times m}$ so that the following inequality holds:*

$$\mathcal{L}_{\mathcal{A}^*}(f_{W^*}) \leq \epsilon$$

*and*

$$\|W_0 - W^*\|_{2,\infty} \leq \frac{K}{m^{3/5}}$$

*The randomness is because of selection of $W_0, b_0, a_0$.*

*Proof.* To prove Theorem B.10, we shall make use of Lemmas B.4, B.9, and Theorem C.1. $f^*$ is the result of Lemma B.4.

As a result of integrating Theorem C.1 with Lemma B.9 and $m \geq \max\{M, \text{poly}(d)\}$, we obtain with prob. $\geq$

$$1 - 1/\exp(\Omega(m^{1/5})) - 1/\exp(\Omega(\sqrt{m/n})),$$

there will be a $W^* \in \mathbb{R}^{d \times m}$ such that

$$\forall x \in \mathcal{X}, |g(x; W^*) - f(x; W^*)| \leq O(\frac{K^2}{m^{1/10}}) \text{ and } |g(x; W^*) - f^*(x)| \leq \epsilon/3. \tag{7}$$

and

$$\|W_0 - W^*\|_{2,\infty} \leq \frac{K}{m^{3/5}}$$

Therefore, for every $\widetilde{x}_i \in \mathcal{B}(x_i, \rho), i \in [n]$,

$$\begin{aligned}
\ell(y_i, f(\widetilde{x}_i; W^*)) &\leq |f(\widetilde{x}_i; W^*) - y_i| \\
&\leq |f^*(\widetilde{x}_i) - y_i| + |g(\widetilde{x}_i; W^*) - f^*(\widetilde{x}_i)| + |f(\widetilde{x}_i; W^*) - g(\widetilde{x}_i; W^*)| \\
&\leq \frac{2\epsilon}{3} + O(\frac{K^2}{m^{1/10}}) \\
&\leq \epsilon
\end{aligned}$$

where the first step follows from the 1-Lipschitz defined in Definition 3.2 , the second step is because of triangle inequality and the third step is due to Eq. (7) and the final step comes from $\epsilon/3 \geq O(\frac{K^2}{m^{1/10}})$. $\text{poly}(d, (n/\epsilon)^{1/\gamma}) \leq m$, given a sufficiently big polynomial. Therefore, we can get $L_{\mathcal{A}^*}(f^*) \leq \epsilon$.

We get $1 - 1/\exp(\Omega(m^{1/5}))$ as the bound of probability by choosing $m \geq n^2$. □

## C  OUR ANALYSIS

We demonstrate the stability of f and g in Section C.1 and the upper bound of the sum of $F_r$ in Section C.2. In Section C.3, we upper bound $\Psi_r$. In Section C.12, we analyze the upper bound of number of activated neurons during initialization. In Section C.4, we define several helper functions $A(x), B(x)$ and $C(x)$. In Section C.5, we first prove the upper bound of $|A(x) - g(x)|$. Then we upper bound $|B(x)|$ in Section C.6. We further upper bound $|C(x)|$ in Section C.7. We can obtain the overall upper bound for $|f(x) - g(x)|$ in Section C.8. We provide some perturbation analysis in Section C.9. We prove the convergence of our algorithm in Section C.10. We analyze the gradient coupling of our algorithm in Section C.11. We upper bound the number of activated neurons for every iteration in Section C.13. We prove the pseudo-network approximation guarantee in Section C.14. We analyze the stability of $S_1$ and $S_2$ in Section C.15.

### C.1  PROOF OF STABILITY OF $f$ AND $g$

Note that when we prove the stability of $f$ and $g$, our definition of $f$ and $g$ is different from Zhang et al. (2020) because of the introduced shifted ReLU operation. With a selection of $\tau = O(\frac{1}{m})$, we prove a different upper bound for $\sup_{x \in \mathcal{X}} |f(x; W) - g(x; W)|$, where $f(x; W) = \sum_{r=1}^m a_{r,0}(\langle w_{r,0} + \Delta w_r, x \rangle + b_{r,0})\Phi_{x,r}$ and $g(x; W) = \sum_{r=1}^m \Phi_{x,r}^{(0)} a_{r,0}\langle \Delta w_r, x \rangle$ compared with Zhang et al. (2020).

**Theorem C.1** (A variation of Theorem 5.1 in Zhang et al. (2020)). *With $K \geq 1$, $\tau = O(\frac{1}{m})$, for every $m \geq \text{poly}(d)$, and every $W \in \mathbb{R}^{d \times m}$ where $\|W_0 - W\|_{2,\infty} \leq \frac{K}{m^{3/5}}$ , with probability at most $1/\exp(\Omega(m^{1/5}))$ over the initialization(See Definition 3.1), we have that*

$$\sup_{x \in \mathcal{X}} |g(x; W) - f(x; W)| \geq O(K^2 \cdot m^{-1/10})$$

*Proof.* We have $\|W - W_0\|_{2,\infty} \leq \frac{K}{m^{3/5}}$, which coincides randomly with the $W_0, a_0, b_0$. The above limitation is enough to get the upper bound of $\|f_W - g_W\|_\infty$ (See Definition 1).

We will take $W$ into consideration now. For simplicity, $f(x; W)$, $g(x; W)$ are written as $f$, $g$ separately. Now, we can get

$$f(x) = \sum_{r=1}^m a_{r,0}(\langle w_{r,0} + \Delta w_r, x \rangle + b_{r,0})\Phi_{x,r}$$

$$g(x) = \sum_{r=1}^m a_{r,0}\langle \Delta w_r, x \rangle \Phi_{x,r}^{(0)}$$

We represent $\Psi_r(x, \eta)$ as follows for any $x \in \mathcal{X}, \eta \in \mathbb{R}_+$ and $r \in [m]$.

$$\Psi_r(x, \eta) := \mathbf{1}[|\langle w_{r,0}, x \rangle + b_{r,0} - \tau| \leq \eta]. \tag{8}$$

With Claim A.10, we can get

$$\Pr[\Psi_r(x, \eta) = 1] < \Pr[|\langle w_{r,0}, x \rangle + b_{r,0}| \leq \eta] \leq O(\eta\sqrt{m}).$$

We define

$$F_r := \mathbf{1}[\Phi_{x,r}^{(0)} \neq \Phi_{x,r}].$$

Now, we can get the bound of the size of $\sum_{r=1}^m F_r$ in Claim C.2. With probability at least $1 - 1/\exp(\Omega(m^{9/10}))$,

$$\sum_{r=1}^m F_r \leq O(K \cdot m^{9/10}), \forall x \in \mathcal{X}.$$

What needs to be done now is union bounding above inequality across each $x \in \mathcal{X}$. Obviously, the issue that $\mathcal{X}$ is not countable still remains. Therefore, across a net of $\mathcal{X}$ with an extremely fine grain, we will get a union bound first and then discuss how $f$ and $g$ have changed as $x \in \mathcal{X}$ as input have changed slightly.

With $\mathcal{X}_1$ as a maximal $\frac{1}{m}$-net of $\mathcal{X}$, we can have that $|\mathcal{X}_1| \leq (\frac{1}{m})^{O(d)}$. By Lemma C.8 and a union bound across $x \in \mathcal{X}_1$, we can know that for $m \geq \Omega(d^3)$,

$$1 - (1/\exp(\Omega(m^{1/5}))) \cdot \exp(O(d \log m)) = 1 - \theta_{1/5},$$

one can obtain the following guarantee:

$$\forall x \in \mathcal{X}_1, \quad |g(x; W) - f(x; W)| \leq O(K^2 \cdot m^{-1/10}) \tag{9}$$

The perturbation analysis comes last. We leverage Lemma C.9 which works for fixed inputs to apply a union bound across $x \in \mathcal{X}_1$. By $m \geq \Omega(d^3)$, for all $x \in \mathcal{X}_1$ and $\mu \in \mathbb{R}^d$ where $\|\mu\|_2 \leq \frac{1}{m}$ and $x + \mu \in \mathcal{X}$, we can have that with probability at most $(1/\exp(\Omega(m^{1/2}))) \exp(O(d \log m)) = 1/\exp(\Omega(m^{1/2}))$, the Eq. (15) and Eq. (14) fail.

Now, with prob. $\geq$

$$1 - 1/\exp(\Omega(m^{1/5})) - 1/\exp(\Omega(m^{1/2})) = 1 - \theta_{1/5}.$$

One can further upper bound $\|g(x; W) - f(x; W)\|_\infty$ by:

$$\|g(x; W) - f(x; W)\|_\infty \leq O(K^2 \cdot m^{-1/10} + m^{-1/5} + K \cdot m^{-1/2} + K \cdot m^{-1})$$
$$= O(K^2 \cdot m^{-1/10})$$

where the first step comes from combining Eq. (9), Eq. (15) and Eq. (14), and applying a union bound, and the second step is due to combining the terms.

The proof is complete.

$\square$

## C.2 Upper bound of sum of $F_r$

The sum of $F_r$ is to be upper bound in this section.

Recall that $F_r = \mathbf{1}[\Phi_{x,r} \neq \Phi_{x,r}^{(0)}]$ and our definitions of $\Phi_{x,r}$ and $\Phi_{x,r}^{(0)}$ are based on the shifted ReLU activation. Note that $\sum_{r=1}^{m} F_r$ represents the number of neurons whose outputs have different signs between the initialized weights $w_{r,0}$ and the updated weights $w_{r,0} + \Delta w_r$. And we obtain a different upper bound of sum of $F_r$ compared with Zhang et al. (2020).

**Claim C.2** (A variation of Claim A.3 in Zhang et al. (2020)). *For each $x \in \mathcal{X}$,*

$$\Pr[\sum_{r=1}^{m} F_r \leq O(K \cdot m^{9/10})] \geq 1 - \theta_{9/10}.$$

*Proof.* With an $x \in \mathcal{X}$ fixed, $\|\Delta W\|_{2,\infty} \leq K/m^{3/5}$ and $\|x\|_2 = 1$, we can get

$$F_r \leq \mathbf{1}[\|\Delta w_r\|_2 \geq |\langle w_{r,0}, x \rangle + b_{r,0} - \tau|]$$
$$\leq \Psi_r(x, K \cdot m^{-3/5}).$$

where the first step comes from the definition of $\Phi_{x,r}^{(0)}$ and $\Phi_{x,r}$(See Definition 1.), and the second step comes from Eq. (8) and $\|\Delta W\|_{2,\infty} \leq K/m^{3/5}$.

With Gaussian anti-concentration from Lemma A.5, we can have that for all $r \in [m]$:

$$\Pr[\Psi_r(x, K/m^{3/5}) = 1] < \Pr[|\langle w_{r,0}, x \rangle + b_{r,0}| \leq K \cdot m^{-3/5}]$$

$$< O(K \cdot m^{-1/10}),$$

By fixing $x$, there will be $m$ Bernoulli random variables that are independent. With Bernstein inequality from Lemma A.9, with probability at most $1/\exp(\Omega(m^{9/10}))$, we attain that

$$\sum_{r=1}^{m} \Psi_r(x, K \cdot m^{-3/5}) \geq O(K \cdot m^{9/10}).$$

We can also have

$$\sum_{r=1}^{m} F_r \leq \sum_{r=1}^{m} \Psi_r(x, K \cdot m^{-3/5}),$$

which implies

$$\sum_{r=1}^{m} F_r \leq O(K \cdot m^{9/10}).$$

The proof is complete now. □

## C.3 UPPER BOUND OF $\Psi_r$

In this section, we want to prove the upper bound of $\Psi_r$.

**Lemma C.3** (Upper bound of $\Psi_r$). *For $x \in \mathcal{X}$, $r \in [m]$, $\eta \in \mathbb{R}_+$ and $\tau > 0$, we define $\Psi_r(x, \eta)$ as follows:*

$$\Psi_r(x, \eta) := \mathbf{1}[|\langle w_{r,0}, x \rangle + b_{r,0} - \tau| \leq \eta].$$

*with probability at least $1 - \theta_{9/10}$*

$$\sum_{r=1}^{m} \Psi_r(x, K/m^{3/5}) \leq O(K \cdot m^{9/10}).$$

*Proof.* We decompose $f$ into three functions $A(x), B(x), C(x)$ in Definition C.4. Then the proof comes from Lemma C.5, Lemma C.6 and Lemma C.7. □

## C.4 DEFINITIONS OF HELP FUNCTIONS

We can split the $f(x)$ into the following three sub-functions for future analysis purpose.

**Definition C.4.** *The following is how $A(x), B(x), C(x)$ are defined:*

$$A(x) := \sum_{r=1}^{m} a_{r,0} \langle \Delta w_r, x \rangle \Phi_{x,r}$$

$$B(x) := \sum_{r=1}^{m} a_{r,0} (\langle w_{r,0}, x \rangle + b_{r,0}) \Phi_{x,r}^{(0)}$$

$$C(x) := \sum_{r=1}^{m} a_{r,0} (\langle w_{r,0}, x \rangle + b_{r,0}) (\Phi_{x,r} - \Phi_{x,r}^{(0)})$$

We will acquire the upper bound of $|A(x) - g(x)|, |B(x)|$ and $|C(x)|$ to upper bound $|g(x) - f(x)|$, because $f(x)$ is the summation of $A(x), B(x)$ and $C(x)$.

Recall that our definitions of $\Phi_{x,r}^{(0)}$ and $\Phi_{x,r}$ are based on shifted ReLU activation with a threshold set as $\tau$, which is different from Zhang et al. (2020).

$$\Phi_{x,r}^{(0)} = \mathbf{1}[\langle w_{r,0}, x \rangle + b_{r,0} \geq \tau]$$
$$\Phi_{x,r} = \mathbf{1}[\langle w_{r,0} + \Delta w_r, x \rangle + b_{r,0} \geq \tau].$$

## C.5 UPPER BOUND OF $|A(x) - g(x)|$

First, we want to obtain the upper bound of $|A(x) - g(x)|$. We obtain a different upper bound based on the shifted ReLU activation compared with Zhang et al. (2020) because our definitions of $\Phi_{x,r}$ and $\Phi_{x,r}^{(0)}$ are different due to the shifted ReLU activation.

**Claim C.5** (A variation of Claim A.5 in Zhang et al. (2020)). *We have that*

$$\Pr[|g(x) - A(x)| \geq O(K^2 \cdot m^{-1/10})] \leq \theta_{9/10}$$

*Proof.* We can easily have that

$$
\begin{aligned}
|A(x) - g(x)| &= |\sum_{r=1}^{m} a_r (\Phi_{x,r} - \Phi_{x,r}^{(0)}) \langle \Delta w_r, x \rangle| \\
&\leq \sum_{r=1}^{m} |a_r| \cdot F_r \cdot |\langle \Delta w_r, x \rangle| \\
&\leq \frac{K}{m} \sum_{r=1}^{m} F_r
\end{aligned}
$$

where the 1st equality is due to the Def. of $A(x)$ and $g(x)$, the 2nd inequality is because $|\Phi_{x,r} - \Phi_{x,r}^{(0)}| \leq F_r$ and the definition of $F_r$, and the last step is because $\|\Delta W\|_{2,\infty} \leq \frac{K}{m^{3/5}}$, $a_r \sim \{\pm \frac{1}{m^{1/5}}\}$. By Claim C.2, we can have that:

$$\Pr[|g(x) - A(x)| \geq O(K^2 \cdot m^{-1/10})] \leq 1/\exp(\Omega(m^{9/10})) = \theta_{9/10}.$$

The proof is complete now.

$\square$

## C.6 UPPER BOUND OF $|B(x)|$

Then we want to upper bound $B(x)$. We obtain a different upper bound $O(1/m^{3/10})$ and different failure probability $\theta_{1/5} = 1/\exp(\Omega(m^{1/5}))$ based on the shifted ReLU activation whose threshold is $\tau$ compared to Zhang et al. (2020).

**Claim C.6** (A variation of Claim A.6 in Zhang et al. (2020)). *With probability at most $\theta_{1/5}$, it gives*

$$|B(x)| \geq O(m^{-1/10})$$

*Proof.* Note that

$$B(x) = \sum_{r=1}^{m} a_{r,0} \sigma_\tau(\langle w_{r,0}, x \rangle + b_{r,0}).$$

With $\sigma_\tau(x) = \mathbf{1}[x > \tau]x$, we can have

$$\sum_{r=1}^{m} \sigma_\tau^2(\langle w_{r,0}, x \rangle + b_{r,0}) \leq \sum_{r=1}^{m} (\langle w_{r,0}, x \rangle + b_{r,0})^2 \tag{10}$$

We can obtain that with probability $\leq \theta_1$,

$$
\begin{aligned}
\sum_{r=1}^{m} \sigma_\tau^2(\langle w_{r,0}, x \rangle + b_{r,0}) &\geq \sum_{r=1}^{m} (\langle w_{r,0}, x \rangle + b_{r,0})^2 \\
&= O(1)
\end{aligned}
$$

where the first step is because of Eq. (10), and the second step is because the random variable $\langle w_{r,0}, x \rangle + b_{r,0}$ is sampled from $\mathcal{N}(0, 2/m)$ independently where $r \in [m]$.

Now, because $a_{r,0}\sigma_\tau(\langle w_{r,0}, x \rangle + b_{r,0})$ for $r \in [m]$ are independence and are Chi-Square random variables(See Definition 3.5.), by using Hoeffding's concentration inequality from Lemma A.6, for some large constant $c > 0$,

$$\Pr[|\sum_{r=1}^{m} a_{r,0}\sigma_\tau(\langle w_{r,0}, x \rangle + b_{r,0})| \geq \frac{c}{m^{1/10}} \mid b_0, W_0]$$

$$\leq \exp(-\Omega(\frac{m^{-1/5}}{\frac{1}{m^{2/5}}\sum_{r=1}^{m} \sigma_\tau^2(\langle w_{r,0}, x \rangle + b_{r,0})})).$$

By utilizing the bound above, with probability at most $1/\exp(\Omega(m^{1/5}))$, we have

$$|B(x)| \geq O(m^{-1/10})$$

The proof is complete. $\qquad\qquad\qquad\qquad\qquad\qquad\qquad\qquad\qquad\qquad\qquad\square$

## C.7 UPPER BOUND OF $|C(x)|$

Finally, we want to obtain a new upper bound $|C(x)|$ based on shifted ReLU in the following Claim C.7. Because our definition of $C(x)$ is different from Zhang et al. (2020), we obtain a different upper bound $O(K^2/m^{3/10})$ and failure probability $\exp(-\Omega(m^{5/6}))$ compared with Zhang et al. (2020).

**Claim C.7** (A variation of Claim A.7 in Zhang et al. (2020))**.** *With $\sigma_\tau$ and probability at most* $\exp(-\Omega(m^{9/10}))$ *, we have*

$$|C(x)| \geq O(K^2 \cdot m^{-1/10}).$$

*Proof.*

$$|C(x)| = \Big| \sum_{r=1}^{m} a_{r,0}(\Phi_{x,r} - \Phi_{x,r}^{(0)})(\langle w_{r,0}, x \rangle + b_{r,0}) \Big|$$

$$\leq \sum_{r=1}^{m} |a_{r,0}||\Phi_{x,r} - \Phi_{x,r}^{(0)}||\langle w_{r,0}, x \rangle + b_{r,0}|$$

$$\leq \frac{1}{m^{1/5}} \sum_{r=1}^{m} |\Phi_{x,r} - \Phi_{x,r}^{(0)}||\langle w_{r,0}, x \rangle + b_{r,0}|$$

where the first step is because of the definition of $C(x)$, the second step is due to triangle inequality, and the third step is due to $|a_{r,0}| = \frac{1}{m^{1/5}}$.

We have that

$$|\Phi_{x,r}^{(0)} - \Phi_{x,r}| \leq F_r$$
$$\leq \Psi_r(x, K \cdot m^{-3/5}). \qquad\qquad (11)$$

And then recall that

$$\Psi_r(x, K \cdot m^{-3/5}) \neq 0 \iff |\langle w_{r,0}, x \rangle + b_{r,0}| \leq K \cdot m^{-3/5}, \qquad (12)$$

which implies that

$$|\langle w_{r,0}, x \rangle + b_{r,0}| \cdot |\Phi_{x,r}^{(0)} - \Phi_{x,r}|$$
$$\leq |\langle w_{r,0}, x \rangle + b_{r,0}| \cdot \Psi_r(x, K \cdot m^{-3/5})$$

$$\leq \frac{K}{m^{3/5}} \cdot \Psi_r(x, K \cdot m^{-3/5})$$

where the first step is becasue of Eq. (11), the final step is due to Eq. (12).

Now we can have,

$$|C(x)| \leq \frac{K}{m} \sum_{r \in [m]} \Psi_r(x, K \cdot m^{-3/5}).$$

As we've already demonstrated, with $\Pr[]$ at most $\theta_{9/10}$,

$$\sum_{r=1}^{m} \Psi_r(x, K \cdot m^{-3/5}) \geq O(K \cdot m^{9/10}).$$

Therefore, with $\Pr[]$ at most $\theta_{9/10}$,

$$|C(x)| \geq O(K^2 \cdot m^{-1/10}).$$

The proof is complete now.

$\square$

## C.8 Upper bound of $|f(x) - g(x)|$

With Claim C.5, Claim C.6 and Claim C.7 in hand, We are prepared to demonstrate that, $|f(x; W) - g(x; W)|$ is tiny for every fixed $x \in \mathcal{X}$ with a high degree of probability. Recall that our definition of $A(x), B(x)$ and $C(x)$ are different from Zhang et al. (2020) and we obtain different upper bounds in Claim C.5, Claim C.6 and Claim C.7. Consequently, we obtain a different upper bound for $|f(x) - g(x)|$ compared with Zhang et al. (2020).

**Lemma C.8** (A variation of Lemma 4 in Zhang et al. (2020)). *For each $x \in \mathcal{X}$, with probability $\leq \theta_{1/5}$,*

$$|f(x; W) - g(x; W)| \geq O(K^2 \cdot m^{-1/10})$$

*Proof.* With Lemma C.5, Lemma C.6 and Lemma C.7 aggregated together and a union bound, for $\forall x \in \mathcal{X}$, with succeed probability

$$1 - \theta_{9/10} - \theta_{1/5} = 1 - \theta_{1/5},$$

it provides

$$|f(x; W) - g(x; W)| \leq |A(x) - g(x)| + |B(x)| + |C(x)|$$
$$\leq O(K^2/m^{1/10}) \tag{13}$$

where the first step is due to triangle inequality and $f(x) = A(x) + B(x) + C(x)$, and the second step is the result of Claim C.5, Claim C.6 and Claim C.7.

This completes the proof. $\square$

## C.9 Perturbation analysis

We do perturbation analysis for $f(x)$ and $g(x)$ with shifted ReLU activation in the following lemma when the perturbation $\mu$ satisfies that $\|\mu\|_2 \leq \frac{1}{m}$ and $x + \mu \in \mathcal{X}$. We obtain a different perturbation upper bound for $|f(x + \mu) - f(x)|$ with the shifted ReLU activation threshold set as $\tau = O(1/m)$ compared with Zhang et al. (2020).

**Lemma C.9** (A variation of Lemma A.8 in Zhang et al. (2020)). *For every $x \in \mathcal{X}_1$, every $\mu \in \mathbb{R}^d$ where $\|\mu\|_2 \leq \frac{1}{m}$ and $x + \mu \in \mathcal{X}$, with $\tau = O(\frac{1}{m})$ and probability at least $1 - 1/\exp(\Omega(m^{1/2}))$, we have*

$$|g(x + \mu; W) - g(x; W)| \leq O(K \cdot m^{-1/2}). \tag{14}$$

*and*

$$|f(x + \mu; W) - f(x; W)| \leq O(K \cdot m^{-1} + 1/m^{1/5}) \tag{15}$$

*Proof.* We define $\mu$ as a small perturbation of $x$ which depends on $a(0) \in \mathbb{R}^d, W_0 \in \mathbb{R}^{m \times d}, b_0 \in \mathbb{R}^d$ arbitrarily and has properties listed in the statement of Lemma.

$$
\begin{aligned}
|f(x + \mu; W) - f(x; W)| &= \Big| \sum_{r=1}^{m} a_{r,0} \Big( \sigma_\tau(\langle w_{r,0} + \Delta w_r, x + \mu \rangle + b_{r,0}) - \sigma_\tau(\langle w_{r,0} + \Delta w_r, x \rangle + b_{r,0}) \Big) \Big| \\
&\leq \sum_{r=1}^{m} |a_{r,0}| (|\langle w_{r,0} + \Delta w_r, \mu \rangle| + \tau) \\
&\leq \frac{1}{m} \sum_{r=1}^{m} |a_{r,0}| \|w_{r,0} + \Delta w_r\|_2 + \sum_{r=1}^{m} |a_{r,0}| \tau \\
&= \frac{1}{m^{1+1/5}} \sum_{r=1}^{m} \|w_{r,0} + \Delta w_r\|_2 + \tau m^{4/5} \\
&\leq \frac{1}{m^{6/5}} \sum_{r=1}^{m} \|w_{r,0}\|_2 + \frac{1}{m^{6/5}} \sum_{r=1}^{m} \|\Delta w_r\|_2 + O(\frac{1}{m^{1/5}}) \\
&\leq \frac{1}{m^{6/5}} \sum_{r=1}^{m} \|w_{r,0}\|_2 + \frac{K}{m} + O(\frac{1}{m^{1/5}}) \\
&\leq O(\frac{1}{m^{1/5}} + \frac{K}{m})
\end{aligned}
$$

where the first step is due to the definition of $f(x; W)$, the second step follows is due to triangle inequality, the third step follows $\|\mu\|_2 \leq \frac{1}{m}$, the fourth step comes from $|a_{r,0}| = \frac{1}{m^{1/5}}$, the fifth step is due to triangle inequality, and the sixth inequality is due to $\sum_{r \in [m]} \|\Delta w_r\|_2 \leq K \cdot m^{1/5}$, and the final step follows that for any $r \in [m]$, with probability $\leq 1/\exp(\Omega(m^2/d))$, we have $\|w_{r,0}\|_2^2 \geq O(1)$.

By concentration (See Lemma A.6.) of the sum of a set of $\chi^2$ random variables which are independent and with union bound over $r \in [m]$ and $m \geq d$,

$$\Pr[O(1) \leq \|W_0\|_{2,\infty}] \leq \theta_1. \tag{16}$$

And then we take $g$ into account.

$$
\begin{aligned}
&|g(x + \mu; W) - g(x; W)| \\
&= \Big| \sum_{r=1}^{m} \mathbf{1}[\langle w_{r,0}, x + \mu \rangle + b_{r,0} \geq \tau] a_{r,0} \langle \Delta w_r, x + \mu \rangle - \sum_{r=1}^{m} \mathbf{1}[\langle w_{r,0}, x \rangle + b_{r,0} \geq \tau] a_{r,0} \langle \Delta w_r, x \rangle \Big| \\
&\leq \frac{1}{m} \sum_{r=1}^{m} |a_{r,0}| \|\Delta w_r\|_2 \\
&\quad + \sum_{r=1}^{m} \Big| \mathbf{1}[\langle w_{r,0}, x + \mu \rangle + b_{r,0} \geq \tau] - \mathbf{1}[\langle w_{r,0}, x \rangle + b_{r,0} \geq \tau] \Big| \cdot |a_{r,0}| \cdot |\langle \Delta w_r, x \rangle| \\
&\leq \frac{K}{m} + \frac{K}{m} \sum_{r=1}^{m} \Big| \mathbf{1}[\langle w_{r,0}, x + \mu \rangle + b_{r,0} \geq \tau] - \mathbf{1}[\langle w_{r,0}, x \rangle + b_{r,0} \geq \tau] \Big|.
\end{aligned}
$$

where the first step is due to the definition of $g(x; W)$, the second step follows from triangle inequality, and the third step is because that $\sum_{r=1}^{m} \|\Delta w_r\|_2 \leq K \cdot m^{1/5}$ and $|a_{r,0}| = \frac{1}{m^{1/5}}$.

About the last sum, from Eq. (16), we have $\|W_0\|_{2,\infty} \leq O(1)$. And the we can attain that

$$\sum_{r=1}^{m} \Big| \mathbf{1}[\langle w_{r,0}, x + \mu \rangle + b_{r,0} \geq \tau] - \mathbf{1}[\langle w_{r,0}, x \rangle + b_{r,0} \geq \tau] \Big| \leq \sum_{r=1}^{m} \Psi_r(x, O(1/m))$$

By Claim A.10, with probability at most $O(\frac{1}{m^{1/2}})$, we can get $\Psi_r(x, O(m^{-1})) = 1$. With $x$ fixed, $\Psi_r(x, O(m^{-1}))$ where $r \in [m]$ can be seen as $m$ independent Bernoulli random variables. Therefore,

by Chernoff bound (See Lemma A.8), with probability at least $1 - 1/\exp(\Omega(m^{1/2}))$, we can have

$$\sum_{r=1}^{m} \Psi_r(x, O(m^{-1})) \leq O(m^{1/2}).$$

The proof of Eq. (14) is finished now.

$\square$

### C.10 Convergence analysis

We first define the $\|\cdot\|_{2,1}$ and $\|\cdot\|_{2,\infty}$ norms in the following definition.

**Definition C.10.** *Let $W \in \mathbb{R}^{d \times m}$ and $w_r \in \mathbb{R}^d$ denote the r-th column of the matrix $W$. We define*

- $\|W\|_{2,1} := \sum_{r=1}^{m} \|w_r\|_2$
- $\|W\|_{2,\infty} := \max_{r \in [m]} \|w_r\|_2$

We define the gradient of real net gradient and pseudo-net gradient as follows:

**Definition C.11.** *The two notions of gradients are represented as convenient notations in the following:*

*Gradient of pseudo-net $\widehat{\nabla}^{(t)}(g) := \nabla_W \mathcal{L}(S^{(t)}, g(W_t)) \in \mathbb{R}^{d \times m}$*

*Gradient of real net $\nabla^{(t)}(f) := \nabla_W \mathcal{L}(S^{(t)}, f(W_t)) \in \mathbb{R}^{d \times m}$*

In the following theorem we analyze the convergence of our neural network $f_W$ with shifted ReLU and choose a different parameter:

$$m \geq \Omega(\max\{n^{300/31}, (Kn/\epsilon)^{24/15}, (K^2/\epsilon)^{300/29}\}),$$
$$\eta = \frac{K}{m^{1/5}\sqrt{T}} = \Theta(m^{-1/5}\epsilon)$$

compared with Zhang et al. (2020).

**Theorem C.12** (A variation of Theorem 4.1 in Zhang et al. (2020)). *For all $K > 0$, and $\epsilon \in (0, 1)$, for every $m$ which is larger than some $\mathrm{poly}(n, K, 1/\epsilon)$. We set*

$$\eta = \Theta(\epsilon m^{-1/5}) \text{ and } T = \Theta(\epsilon^{-2} K^2),$$

*Running Algorithm 1, for every $W^* \in \mathbb{R}^{d \times m}$ with $\|W_0 - W^*\|_{2,\infty} \leq \frac{K}{m^{3/5}}$, there will be weights $\{W_t\}_{t=1}^{T} \in \mathbb{R}^{d \times m}$ such that*

$$\mathcal{L}_{\mathcal{A}^*}(f_{W^*}) + \epsilon \geq \frac{1}{T} \sum_{t=1}^{T} \mathcal{L}_{\mathcal{A}}(f_{W_t})$$

*The succeed probability is $1 - \theta_{1/5}$. The randomness is coming from initialized states (See Definition 3.1).*

*Proof.* As for $T$ and $\eta$, we will give them value in the proof later. We define various distances as follows

$$D_{\max} := \max_{t \in [T]} \|W_t - W_0\|_{2,\infty} \tag{17}$$

and

$$D_{W^*} := \|W^* - W_0\|_{2,\infty} \tag{18}$$

where $D_{W^*} = O(\frac{K}{m^{3/5}})$.

By Lemma C.14, with high probability $D_{\max}$ and $D_{W^*}$ will be coupled, if $W_t$ remains near initiation $(D_{\max} \leq m^{-15/24})$.

$$\|\widehat{\nabla}^{(t)}(g) - \nabla^{(t)}(f)\|_{2,1} \leq O(nm^{27/50})$$

**Remark C.13.** *We assume $D_{\max} \le m^{-15/24}$ first. Finally, we will set $\eta, T$ and $m$ to the appropriate values to ensure that it occurs.*

We can bound the size of gradient for all $r \in [m]$:

$$
\|\nabla^{(t)}(f)_r\|_2 \le (\frac{1}{n}\sum_{i=1}^{n}\sigma_\tau(\langle w_{r,t}, x_i\rangle + b_{r,0}) \cdot \|\widetilde{x}_i\|_2) \cdot |a_r|
$$
$$
\le \frac{1}{m^{1/5}} \tag{19}
$$

where the first step is because the loss is 1-Lipschitz, and the second step is due to $|a_r| = 1/m^{1/5}$ and $\sigma_\tau(\langle w_{r,t}, x_i\rangle + b_{r,0}) \cdot \|\widetilde{x}_i\|_2 \le O(1)$.

The $\mathcal{L}(S, (g(W))$ is a convex function with respect to $W \in \mathbb{R}^{d \times m}$ because $g$ is linear with respect to $W \in \mathbb{R}^{d \times m}$.

We express the inner product of two two identically sized matrices $B$ and $C$ as

$$
\langle B, C \rangle := \text{tr}[B^\top C]
$$

And then we have:

$$
\mathcal{L}(S^{(t)}, g(W_t)) - \mathcal{L}(S^{(t)}, g(W^*))
$$
$$
\le \langle \widehat{\nabla}^{(t)}(g) - \nabla^{(t)}(f), W_t - W^* \rangle + \langle \nabla^{(t)}(f), W_t - W^* \rangle
$$
$$
\le \|\widehat{\nabla}^{(t)}(g) - \nabla^{(t)}(f)\|_{2,1} \cdot \|W_t - W^*\|_{2,\infty} + \langle \nabla^{(t)}(f), W_t - W^* \rangle \tag{20}
$$

where the first step is due to triangle inequality, and the second step is because the inner product and the definition of $\|\cdot\|_{2,1}$ and $\|\cdot\|_{2,\infty}$.

For simplicity, we define $\alpha(t) \in \mathbb{R}$ and $\beta(t) \in \mathbb{R}$ as follows:

$$
\alpha(t) := \langle \nabla^{(t)}(f), W_t - W^* \rangle
$$
$$
\beta(t) := \|\widehat{\nabla}^{(t)}(g) - \nabla^{(t)}(f)\|_{2,1} \cdot \|W_t - W^*\|_{2,\infty}
$$

Therefore, we have:

$$
\mathcal{L}(S^{(t)}, g(W_t)) - \mathcal{L}(S^{(t)}, g(W^*)) \le \alpha(t) + \beta(t)
$$

$\alpha(t)$ and $\beta(t)$ terms will be dealt with separately.

As for $\alpha(t)$, we can have:

$$
\|W_{t+1} - W^*\|_F^2 = \|W_t - \eta\nabla^{(t)}(f) - W^*\|_F^2
$$
$$
= \|W_t - W^*\|_F^2 + \eta^2\|\nabla^{(t)}(f)\|_F^2 - 2\eta\alpha(t)
$$

where the first step us due to $W_{t+1} = W_t - \eta\nabla^{(t)}(f)$, and the second step is because of rearranging the terms.

And then by reorganizing the terms, we can have

$$
\alpha(t) \le \frac{\eta}{2}\|\nabla^{(t)}(f)\|_F^2 + \frac{1}{2\eta} \cdot (\|W_t - W^*\|_F^2 - \|W_{T+1} - W^*\|_F^2)
$$

By summing over $t$, we have

$$
\sum_{t=1}^{T}\alpha(t) \le \frac{\eta}{2}\sum_{t=1}^{T}\|\nabla^{(t)}(f)\|_F^2 + \frac{1}{2\eta} \cdot (\|W_0 - W^*\|_F^2 - \|W_{T+1} - W^*\|_F^2)
$$
$$
\le \frac{\eta m^{1/5}}{2}T + \frac{mD_{W^*}^2}{2\eta} \tag{21}
$$

where the first inequality is due to the upper bound of $\alpha(t)$, and the second inequality is because $\|W^* - W_0\|_F^2 \le m \cdot \|W^* - W_0\|_{2,\infty} = m D_{W^*}^2$ and $\|\nabla^{(t)}(f)\|_F^2 \le \sum_{r=1}^m \|\nabla_r^{(t)}\|_2^2 \le m^{1/5}$.

For the $\beta(t)$'s,

$$\begin{aligned}
\beta(t) &\le \|W_t - W^*\|_{2,\infty} \cdot O(nm^{27/50}) \\
&\le (D_{W^*} + D_{\max}) \cdot O(nm^{27/50})
\end{aligned} \tag{22}$$

where the 1st inequality is because Lemma C.14, and the 2nd inequality is due to triangle inequality and the definition of $D_{\max}$ and $D_{W^*}$ in Eq. (17) and Eq. (18).

Additionally, we may get the bound of $D_{\max}$'s size using the bound of gradients.

$$\begin{aligned}
D_{\max} &= \max_{t \in [T]} \|W_0 - W_t\|_{2,\infty} \\
&\le \sum_{t=1}^T \eta \max_{r \in [m]} \|\nabla^{(t)}(f)_r\|_2 \\
&\le \frac{\eta T}{m^{1/5}}
\end{aligned}$$

where the first equality is due to the value of $D_{\max}$, the second inequality replies on accumulating $T$ iterations, and the final inequality is due to Eq. (19).

Combining it with the preexisting condition $D_{W^*} = O(\frac{K}{m^{4/5}})$, we arrive to the following:

$$\begin{aligned}
&\sum_{t=1}^T \mathcal{L}(S^{(t)}, g(W_t)) - \sum_{t=1}^T \mathcal{L}(S^{(t)}, g(W^*)) \\
&\le \sum_{t=1}^T \alpha(t) + \sum_{t=1}^T \beta(t) \\
&\le O(1)(m^{1/5}\eta T + \frac{K^2}{m^{1/5}\eta} + \eta T^2 nm^{31/150} + \frac{\eta KTn}{m^{29/150}})
\end{aligned}$$

where the first step is due to Eq. (20), and the second step is due to the upper bound in Eq. (21) and Eq. (22).

Then,

$$\frac{1}{T}\sum_{t\in[T]} \mathcal{L}(S^{(t)}, g(W_t)) - \frac{1}{T}\sum_{t\in[T]} \mathcal{L}(S^{(t)}, g(W^*)) \le O(\epsilon).$$

The following values were chosen for the hyper-parameters $\eta, m$, and $T$:

$$\eta = \frac{K}{m^{1/5}\sqrt{T}} = \Theta(m^{-1/5}\epsilon),$$

$$m \ge \Omega(\max\{n^{300/31}, (Rn/\epsilon)^{24/15}, (K^2/\epsilon)^{300/29}\}),$$

$$T = \Theta(\epsilon^{-2}K^2)$$

where $m$ is required to satisfy $\eta T^2 nm^{31/150} + \frac{\eta KTn}{m^{29/150}} \le O(\epsilon)$, $D_{\max} \le m^{-15/24}$ and to fulfill the prerequisite for using the Theorem C.1, we have for all $t \in [T]$ that

$$\sup_{x \in \mathcal{X}} |g(x; W_t) - f(x; W_t)| \le O(\epsilon)$$

Therefore, we have

$$\frac{1}{T}\sum_{t=1}^{T}\mathcal{L}(S^{(t)}, f_{W_t}) - \frac{1}{T}\sum_{t=1}^{T}\mathcal{L}(S^{(t)}, f_{W^*}) \leq c \cdot \epsilon$$

where $c \in \mathbb{R}_+$ is a large constant. With $\mathcal{L}(S^{(t)}, f_{W_t}) = \mathcal{L}_{\mathcal{A}}(f_{W_t})$ and $\mathcal{L}(S^{(t)}, f_{W^*}) \leq \mathcal{L}_{\mathcal{A}^*}(f_{W^*})$, if we replace $\epsilon$ with $\frac{\epsilon}{c}$, the proof will hold for all $\epsilon > 0$. We finish the proof now. $\qquad\square$

### C.11 GRADIENT COUPLING

In this section, we show that $\|\widehat{\nabla}^{(t)}(g) - \nabla^{(t)}(f)\|_{2,1}$ ($\|\cdot\|_{2,1}$ is from Definition C.10) can be upper bounded with high probability for all iterations $t$. Due to the new shifted ReLU activation with threshold $\tau$, the new upper bound is $O(nm^{27/50})$ which is different from Zhang et al. (2020).

**Lemma C.14** (A variation of Lemma A.10 in Zhang et al. (2020)). *For every* $t \in [T]$ *let* $\|W_t - W_0\|_{2,\infty} \leq O(m^{-15/24})$. *Then*

$$\Pr[\|\widehat{\nabla}^{(t)}(g) - \nabla^{(t)}(f)\|_{2,1} \leq O(nm^{27/50})] \geq 1 - \theta_{1/5}.$$

*Proof.* For every $t \in [T]$, by Claim C.16 and $D_{\max} = \|W_t - W_0\|_{2,\infty}$, we have,

$$\Pr[\sum_{r=1}^{m}\mathbf{1}[\nabla_r^{(t)} \neq \widehat{\nabla}_r^{(t)}] \leq O(nm^{7/8})] \geq 1 - \theta_{1/5}. \tag{23}$$

For $r \in [m]$ where $\nabla_r^{(t)} \neq \widehat{\nabla}_r^{(t)}$,

$$\|\widehat{\nabla}_r^{(t)} - \nabla_r^{(t)}\|_2 \leq |a_r|\frac{1}{n}\sum_{i=1}^{n}|\|\widetilde{x}_i\|_2\,\mathbf{1}[\langle w_{r,t}, \widetilde{x}_i\rangle + b_{r,0} \geq \tau] - \mathbf{1}[\langle w_{r,0}, x_i\rangle + b_{r,0} \geq \tau]|$$

$$\leq \frac{1}{m^{1/5}}\frac{1}{n}\sum_{i=1}^{n}|\mathbf{1}[\langle w_{r,t}, \widetilde{x}_i\rangle + b_{r,0} \geq \tau] - \mathbf{1}[\langle w_{r,0}, x_i\rangle + b_{r,0} \geq \tau]|$$

$$\leq \frac{1}{m^{1/5}} \tag{24}$$

where the first step is because that the loss is 1-Lipschitz, and the second step is because that $a_r \in \{-\frac{1}{m^{1/5}}, \frac{1}{m^{1/5}}\}$, and the third step is because that $|\mathbf{1}[\langle w_{r,t}, \widetilde{x}_i\rangle + b_{r,0} \geq \tau] - \mathbf{1}[\langle w_{r,0}, x_i\rangle + b_{r,0} \geq \tau]| \leq 1$.

Thus, we conclude

$$\|\widehat{\nabla}^{(t)}(g) - \nabla^{(t)}(f)\|_{2,1} = \sum_{r=1}^{m}\|\widehat{\nabla}_r^{(t)} - \nabla_r^{(t)}\|_2$$

$$\leq \frac{1}{m^{1/5}} \cdot O(nm^{7/8})$$

$$= O(nm^{27/50})$$

where the first step is due to the definition of $\|\cdot\|_{2,1}$, the second step is due to Eq. (23) and Eq. (24), and the final step is the result of merging the terms.

$\qquad\square$

### C.12 UPPER BOUND OF NUMBER OF ACTIVATED NEURONS AT INITIALIZATION

We prove the upper bound of number of activated neurons with shifted ReLU activation whose threshold is $\tau$ at initialization. We will use the below lemma to compute the time cost per training iteration.

**Lemma C.15** (Restatement of Lemma 4.5, Number of activated neurons during initialization). *Let $c_Q \in (0, 1)$ denote a fixed constant. Let $Q_0 = 2m \cdot \exp(-(\frac{K^2}{4m^{1/5}})) = O(m^{c_Q})$. For every $\eta \in \mathbb{R}_+$, $x \in \mathcal{X}$ and $r \in [m]$, $\Psi_r(x, \eta)$ is defined as follows:*

$$\Psi_r(x, \eta) := \mathbf{1}[|\langle w_{r,0}, x \rangle + b_{r,0}| \geq \eta].$$

*with succeed probability $\geq 1 - n/\exp(\Omega(m \cdot \exp(-(\frac{K^2}{4m^{1/5}}))))$*

$$\sum_{r \in [m]} \Psi_r(x, K/m^{3/5}) \leq Q_0.$$

*Proof.* Fix $x$, $\langle w_{r,0}, x \rangle + b_{r,0} \sim \mathcal{N}(0, \frac{2}{m})$, due to the reason that $b_0$ and $W_0$ are random variables from $\mathcal{N}(0, \frac{1}{m})$ independently. By Gaussian tail bounds and replacing $\langle w_{r,0}, x \rangle + b_{r,0}$ with $z$, the probability that each initial neuron is activated is

$$\Pr[\langle w_{r,0}, x \rangle + b_{r,0} \geq K \cdot m^{-3/5}] = \Pr_{z \sim \mathcal{N}(0, \frac{2}{m})}[z \geq K \cdot m^{-3/5}]$$

$$\leq \exp(-(\frac{K^2}{4m^{1/5}}))$$

By using $\mathbf{1}_{r \in Q_{0,i}}$, we have

$$\mathbb{E}[\mathbf{1}_{r \in Q_{0,i}}] \leq \exp(-(\frac{K^2}{4m^{1/5}}))$$

By Bernstein inequality from Lemma A.9, with $\eta > 0$

$$\Pr[Q_{i,0} > \eta + t] \leq \exp(-\frac{t^2/2}{\eta + t/3}), \forall t > 0$$

where $\eta := m \cdot \exp(-(\frac{K^2}{4m^{1/5}}))$.

By choosing $t = \eta$, we can have:

$$\Pr[Q_{i,0} > 2\eta] \leq \exp(-3\eta/8)$$

By applying union bound across $i \in [n]$, we can attain that for each $x_i \in \mathcal{X}$ with at least probability

$$1 - n \cdot 1/\exp(\Omega(m \cdot \exp(-(\frac{K^2}{4m^{1/5}})))),$$

the upper bound of $Q_{i,0}$ (See definition 3.9.) is $2m \cdot \exp(-(\frac{K^2}{4m^{1/5}}))$. □

### C.13 UPPER BOUND OF THE SIZE OF ACTIVATED NEURON SET

We have obtained the upper bound of the size of activated neuron set at initialization. In this section, we want to upper bound the size of neuron set which has different signs compared to the neuron weights at initialization under shifted ReLU activation per training iteration. When computing the sign, we need to take the activation threshold $\tau$ into account compared with Zhang et al. (2020).

**Claim C.16** (A variation of Claim A.11 in Zhang et al. (2020)). *For any $\|\Delta w_r\|_2 \leq m^{-15/24}$ and any subset $\{x_i\}_1^n \subseteq \mathcal{X}$. Then we have*

$$\sum_{r=1}^m \mathbf{1}[\exists i \in [n], \ \text{sgn}(\langle x_i, w_{r,0} + \Delta w_r \rangle + b_{r,0} - \tau) \neq \text{sgn}(\langle x_i, w_{r,0} \rangle + b_{r,0} - \tau)] < O(nm^{7/8})$$

*and*

$$\forall i \in [m], \mathbf{1}[\exists i \in [n], \ \text{sgn}(\langle x_i, w_{r,0} + \Delta w_r \rangle + b_{r,0} - \tau) \neq \text{sgn}(\langle x_i, w_{r,0} \rangle + b_{r,0} - \tau)] < O(nm^{-1/8})$$

*The succeed probability is $1 - \theta_{1/5}$.*

*The randomness is from initialized states (See Definition 3.1).*

*Proof.* We first give a proof of Claim C.16 with $n$ points in the set fixed. At last we get the final result by using a union bound across all the sets.

With $\{x_1, \ldots, x_n\} \subseteq \mathcal{X}$ where $x_i$'s are fixed, we have the following definition:

$$C_r := \mathbf{1}[\exists i \in [n], \ \mathrm{sgn}(\langle w_{r,t}, x_i\rangle + b_{r,0} - \tau) \neq \mathrm{sgn}(\langle w_{r,0}, x_i\rangle + b_{r,0} - \tau)]$$

Now we will focus on bounding the size of $\sum_{r=1}^m C_r$. By Claim A.10, for $x_i \in \mathcal{X}$ we have that

$$\Pr[|\langle w_{r,0}, x_i\rangle + b_{r,0} - \tau| \leq m^{-15/24}] < \Pr[|\langle w_{r,0}, x_i\rangle + b_{r,0}| \leq \frac{1}{m^{15/24}}]$$
$$\leq O(\frac{1}{m^{1/8}})$$

By a union bound across $i \in [n]$, we have

$$\Pr[\exists i \in [n] \text{ s.t. } |\langle w_{r,0}, x_i\rangle + b_{r,0} - \tau| \leq m^{-15/24}] \leq O(\frac{n}{m^{1/8}})$$

such that

$$\Pr[C_r = 1] \leq \Pr[\exists i \in [n] \text{ s.t. } |\langle w_{r,0}, x_i\rangle + b_{r,0} \geq \tau| \leq \frac{1}{m^{15/24}}]$$
$$< O(\frac{n}{m^{1/8}})$$

With $x_i \in \mathcal{X}$'s fixed, $C_r$'s will be $m$ Bernoulli random variables which are independent where $r \in [m]$. Therefore, by Chernoff bound from Lemma A.8,

$$\Pr[\sum_{r \in [m]} C_r \geq O(m^{7/8}n)] \leq 1/\exp(\Omega(m^{7/8}n)).$$

We will only amplify by just $\exp(O(nd\log m))$ on the failure probability with a big enough $m$ when over product space $\otimes^n \mathcal{X}$, we assume a union bound over a $\frac{1}{m}$-net .

$\square$

**Remark C.17.** *In each training iteration we only use $Q_0 + O(nm^{7/8})$ activated neurons to do the computation to save training cost. We will use the second upper bound to compute the time eventually. $Q_0$ is the activated neuron set size during initialization.*

## C.14    PROOF OF PSEUDO-NETWORK APPROXIMATION

We will first demonstrate how individual components of $f^*$ may be approximated by pseudo-networks and combine them to create a sizable pseudo-network by which $f^*$ is approximated.

**Lemma C.18** (A variation of Lemma A.16 in Zhang et al. (2020)). *Let $q : \mathbb{R} \to \mathbb{R}$ univariate polynomial, $i \in [n]$, $c_1$ be a large constant, $\epsilon_3 \in (0, \frac{1}{\mathcal{C}(q)})$ and $k \geq c_1 \frac{d}{\epsilon_3^2}\mathcal{C}^2(q, \epsilon_3)$. For every $r \in [k]$, with independent random variables $w_{r,0}$, $b_{r,0}$ and $a_{r,0}$, where $w_{r,0}$ is a d-dimentional standard Gaussian random variable, $b_{r,0}$ is a standard normal variable , and $a_{r,0} \sim \{-\frac{1}{m^{1/5}}, +\frac{1}{m^{1/5}}\}$ uniformly at random. With $\tau = O(k^{-1/2})$ and probability at least $1 - 1/\exp(\Omega(k^{1/2}))$, there will be $\Delta W^{(i)} \in \mathbb{R}^{d \times k}$ such that*

$$\forall x \in \mathcal{X}, |\sum_{r=1}^k \alpha_{r,0}\langle\Delta w_r(i), x\rangle \mathbf{1}[\langle w_{r,0}, x\rangle + b_{r,0} \geq \tau] - y_i q(\langle x_i, x\rangle)| \leq 3\epsilon_3$$

*and*

$$\|\Delta W^{(i)}\|_{2,\infty} \leq O(\frac{\mathcal{C}(q, \epsilon_3)}{k} \cdot m^{1/5})$$

*Proof.* Notice that $\langle w_r, x \rangle + b_r \sim \mathcal{N}(0,2)$, and then by claim A.10, we can have $\Pr[\mathbf{1}[\langle w, x \rangle + b < \tau]] = O(\tau)$. And then there will a function defined as $h : \mathbb{R}^2 \to [-\mathcal{C}(q,\epsilon_3), \mathcal{C}(q,\epsilon_3)]$ which satisfies that:

$$\left| \mathbb{E}_{b \sim \mathcal{N}(0,1), w \sim \mathcal{N}(0,I_d)} [\mathbf{1}[\langle w, x \rangle + \beta < \tau] \, h(\langle x_i, w \rangle, b)] \right| \leq O(\tau)\mathcal{C}(q,\epsilon_3) \tag{25}$$

We use Lemma B.3 by setting $\phi(z) = y_i q(z)$ and $\epsilon_1 = \epsilon_3$. Note that $|y_i| \leq O(1)$, $\phi$ will have an equal complexity to $q$, with some constant. Therefore, we have a function $h : \mathbb{R}^2 \to [-\mathcal{C}(q,\epsilon_3), \mathcal{C}(q,\epsilon_3)]$ s. t. for every $x$ in set $\mathcal{X}$

$$\left| \mathbb{E}_{b \sim \mathcal{N}(0,1), w \sim \mathcal{N}(0,I_d)} [h(\langle x_i, w \rangle, b) \, \mathbf{1}[\langle w, x \rangle + b \geq \tau]] - y_i q(\langle x_i, x \rangle) \right|$$

$$\leq \left| \mathbb{E}_{b \sim \mathcal{N}(0,1), w \sim \mathcal{N}(0,I_d)} [h(\langle x_i, w \rangle, b) \, \mathbf{1}[\langle w, x \rangle + b \geq 0]] - y_i q(\langle x_i, x \rangle) \right|$$

$$+ \left| \mathbb{E}_{b \sim \mathcal{N}(0,1), w \sim \mathcal{N}(0,I_d)} [h(\langle x_i, w \rangle, b) \, \mathbf{1}[\langle w, x \rangle + b < \tau]] \right|$$

$$\leq \epsilon_3 + O(\tau)\mathcal{C}(q,\epsilon_3)$$

$$\leq \epsilon_3 + O\left(\frac{\mathcal{C}(q,\epsilon_3)}{k^{1/2}}\right) \tag{26}$$

where the first step is due to triangle inequality, and the second step is because of Lemma B.3 and Eq.(25), the last step comes from $\tau = O(k^{-1/2})$.

With an $x \in \mathcal{X}$ fixed and by Hoeffding's inequality (Lemma A.6), with probability at least $1 - 1/\exp(\Omega(\frac{\epsilon_3^2 k}{\mathcal{C}^2(q,\epsilon_3)}))$, we have

$$\left| \frac{1}{k} \sum_{r=1}^{k} \mathbf{1}[\langle w_{r,0}, x \rangle + b_{r,0} \geq \tau] h(\langle w_{r,0}, x_i \rangle, b_{r,0}) - \right.$$

$$\left. \mathbb{E}_{b \sim \mathcal{N}(0,1), w \sim \mathcal{N}(0,I_d)} [\mathbf{1}[\langle w, x \rangle + b \geq \tau]] \, h(\langle w, x_i \rangle, b)] \right| \leq \epsilon_3 \tag{27}$$

Let

$$\Delta w_r(i) = \frac{1}{\alpha_{r,0}} \frac{1}{k} \cdot 2h(\langle w_{r,0}, x_i \rangle, b_{r,0}) \cdot \widehat{e}.$$

where $\widehat{e}$ is a vector whose only the last element is 1 and otherwise is 0, we can obtain the following upper bound:

$$\|\Delta W^{(i)}\|_{2,\infty} \leq O\left(\frac{\mathcal{C}(q,\epsilon_3)}{k} \cdot m^{1/5}\right)$$

For every $x \in \mathcal{X}$, because $x_d = 1/2$, with probability at most $\exp(-\Omega(\frac{k\epsilon_3^2}{\mathcal{C}^2(q,\epsilon_3)}))$, we have

$$\left| \sum_{r=1}^{k} \mathbf{1}[\langle w_{r,0}, x \rangle + b_{r,0} \geq \tau]\alpha_{r,0}\langle \Delta w_r(i), x \rangle - \right.$$

$$\left. \mathbb{E}_{b \sim \mathcal{N}(0,1), w \sim \mathcal{N}(0,I_d)} [h(\langle x_i, w \rangle, b) \, \mathbf{1}[\langle w, x \rangle + b \geq \tau]] \right| \geq \epsilon_3 \tag{28}$$

With $\mathcal{X}_1$ as a maximal $\frac{1}{k^c}$-net of $\mathcal{X}$ and $c \in \mathbb{R}_+$ as a large enough constant , we all understand that $|\mathcal{X}_1| \leq (\frac{1}{k})^{O(d)}$. With a union bound over $\mathcal{X}_1$ for Eq. (28) and $c_1$ as a significant constant, we have that for $k \geq c_1 \frac{d}{\epsilon_3^2}\mathcal{C}^2(q,\epsilon_3))$,

$$\Pr\left[ \left| \sum_{r=1}^{k} \mathbf{1}[\langle w_{r,0}, x \rangle + b_{r,0} \geq \tau]a_{r,0}\langle \Delta w_{r,i}, x \rangle \right. \right.$$

$$- \mathop{\mathbb{E}}_{b \sim \mathcal{N}(0,1), w \sim \mathcal{N}(0,I_d)} [h(\langle w, x_i \rangle, b) \ \mathbf{1}[\langle w, x \rangle + b \geq \tau]] \Bigg| > \epsilon_3, \ \forall x \in \mathcal{X}_1 \Bigg]$$

$$\leq (1/\exp(\Omega(\frac{k\epsilon_3^2}{\mathcal{C}^2(q,\epsilon_3)}))) \exp(O(d \log k))$$

$$= 1/\exp(\Omega(\frac{k\epsilon_3^2}{\mathcal{C}^2(q,\epsilon_3)})) \tag{29}$$

where the first step is due to Eq.(27), the second step is because of adding terms, and the last step demonstrate that for each $x \in \mathcal{X}_1$, with a high probability, if $x$ is adjusted by no more than $\frac{1}{k^c}$ in $\ell_2$, the LHS of Eq. (28) only slightly changes.

After showing that a given $x \in \mathcal{X}$ is stable, a union bound will be performed.

Indeed, with a large enough $c$ and combining Eq. (26), (29) and Claim C.19, with probability at least $1 - 1/\exp(\Omega(k^{1/2})) - 1/\exp(\Omega(\frac{k\epsilon_3^2}{\mathcal{C}^2(q,\epsilon_3)}))$, we attain that

$$\forall x \in \mathcal{X}, |\sum_{r=1}^{k} a_{r,0} \langle \Delta w_{r,i}, x \rangle \mathbf{1}[\langle w_{r,0}, x \rangle + b_{r,0} \geq \tau] - y_i q(\langle x_i, x \rangle)| \leq O(k^{-1/2} \cdot \mathcal{C}(q, \epsilon_3)) + 2\epsilon_3$$

since $k \geq c_1 \frac{d}{\epsilon_3^2} \mathcal{C}^2(q, \epsilon_3)$ for a large constant $c_1$, the proof is complete. $\qquad\square$

### C.15 STABILITY OF $S_1$ AND $S_2$

In this section, we'll talk about the stability of $S_1$ and $S_2$ based on the shifted ReLU activation with a threshold set as $\tau$. When we bound $S_1$ and $S_2$, we need to bound the sum of $k$ independent Bernoulli random variables $\mathbf{1}[|\langle w_{r,0}, x \rangle + b_{r,0} - \tau| \leq \frac{1}{\sqrt{k}}]$ which is different from Zhang et al. (2020).

**Claim C.19** (A variation of Claim A.17 in Zhang et al. (2020)). *For $\tau > 0$ and every $x \in \mathcal{X}_1$, let $c$ be large enough and $\mu \in \mathbb{R}^d$ such that $\|\mu\|_2 \leq \frac{1}{k^c}$ and $x + \mu \in \mathcal{X}$. With independent random variables $w_{r,0}$, $b_{r,0}$ and $a_{r,0}$, where $w_{r,0}$ is a d-dimentional standard Gaussian random variable, $b_{r,0}$ is a standard normal variable, and probability $\geq 1 - 1/\exp(\Omega(k^{1/2}))$, we have*

$$S_1 := \left| \sum_{r=1}^{k} \alpha_{r,0} \langle \Delta w_r(i), x + \mu \rangle \mathbf{1}[\langle w_{r,0}, x + \mu \rangle + b_{r,0} \geq \tau] - \sum_{r=1}^{k} \alpha_{r,0} \langle \Delta w_r(i), x \rangle \mathbf{1}[\langle w_{r,0}, x \rangle + b_{r,0} \geq \tau] \right|$$

$$\leq O(k^{-1/2} \cdot \mathcal{C}(q, \epsilon_3))$$

*and*

$$S_2 := \Bigg| \mathop{\mathbb{E}}_{b \sim \mathcal{N}(0,1), w \sim \mathcal{N}(0,I_d)} [ h(\langle w, x_i \rangle, b) \ \mathbf{1}[\langle w, x + \mu \rangle + b \geq \tau]]]$$

$$- \mathop{\mathbb{E}}_{b \sim \mathcal{N}(0,1), w \sim \mathcal{N}(0,I_d)} [ h(\langle w, x_i \rangle, b) \ \mathbf{1}[\langle w, x \rangle + b \geq \tau]] \Bigg|$$

$$\leq O(k^{-1/2} \cdot \mathcal{C}(q, \epsilon_3))$$

*Proof.* The upper bound of $D_1$ will be given first. Note that $\Delta W_{rj}^{(i)} = 0$ for $j \leq d - 1$ based on how we generated $\Delta W(i)$. And $\langle \Delta w_r(i), \mu \rangle = 0$ when $v_d = 0$.

With conditions as follows:

$$|\alpha_{r,0}| = \frac{1}{k^{1/5}} \ \text{ and } \ \|\Delta W^{(i)}\|_{2,\infty} \leq O(\frac{\mathcal{C}(q, \epsilon_3) m^{1/5}}{k}),$$

we will have

$$S_1 \leq O(\frac{\mathcal{C}(q, \epsilon_3)}{k}) \sum_{r=1}^{k} \Big| \mathbf{1}[\langle w_{r,0}, x + \mu \rangle + b_{r,0} \geq \tau] - \mathbf{1}[\langle w_{r,0}, x \rangle + b_{r,0} \geq \tau] \Big|$$

$$\leq O(\frac{\mathcal{C}(q, \epsilon_3)}{k}) \sum_{r=1}^{k} \mathbf{1}[\mathrm{sgn}(\langle w_{r,0}, x + \mu \rangle + b_{r,0} - \tau) \neq \mathrm{sgn}(\langle w_{r,0}, x \rangle + b_{r,0} - \tau)]$$

$$\leq O(\frac{\mathcal{C}(q, \epsilon_3)}{k}) \sum_{r=1}^{k} (\mathbf{1}[|\langle w_{r,0}, x \rangle + b_{r,0} - \tau| \leq k^{-1/2}] + \mathbf{1}[\|w_{r,0}\|_2 > c_2 \cdot k^{1/2}]).$$

where the first step is due to $|h(\cdot)| \leq \mathcal{C}(q, \epsilon_3)$, the second step is the result of the value of $\left| \mathbf{1}[\langle w_{r,0}, x + \mu \rangle + b_{r,0} \geq \tau] - \mathbf{1}[\langle w_{r,0}, x \rangle + b_{r,0} \geq \tau] \right|$ is determined by the number of differences between the sign of $\langle w_{r,0}, x + \mu \rangle + b_{r,0}$ and $\langle w_{r,0}, x \rangle + b_{r,0}$, and the third step follows that only when $\langle w_{r,0}, x + \mu \rangle + b_{r,0}$ and $\langle w_{r,0}, x \rangle + b_{r,0}$ have different sign, $\mathbf{1}[\langle w_{r,0}, x + \mu \rangle + b_{r,0}) \neq \mathrm{sgn}(\langle w_{r,0}, x \rangle + b_{r,0})]$ is 1 which is decided by $w_{r,0}$ and $\mu$, where $\|\mu\|_2 \leq \frac{1}{k^c}$, $w \sim \mathcal{N}(0, I_d)$ and $\mathbf{1}[\|w_{r,0}\|_2 > c_2 k^{1/2}] \geq 0$.

We can choose a large enough constant $c_2$ if an sufficiently large constant $c$ is chosen.

And then we demonstrate that with probability at least $1 - 1/\exp(\Omega(k))$, for every $r \in [k]$, $\|w_{r,0}\|_2 \leq O(k^{1/2})$.

By the concentration property of sum of a set of independent Chi-Square random variables from Lemma B.2 and Lemma A.7, for all $r \in [k]$, with probability $\leq 1/\exp(\Omega(k^2/d))$, we have $\|w_{r,0}\|_2^2 \geq O(k)$. By applying union bound over all $r \in [k]$ we can complete the proof with $k \geq d$.

Therefore, by selecting $c_2$ correctly, we can obtain the following bound:

$$S_1 \leq O(k^{-1} \cdot \mathcal{C}(q, \epsilon_3)) \sum_{r=1}^{k} \mathbf{1}[|\langle w_{r,0}, x \rangle + b_{r,0} - \tau| \leq k^{-1/2}] \tag{30}$$

The succeed probability is $\geq 1 - 1/\exp(\Omega(k))$.

Now, $\mathbf{1}[|\langle w_{r,0}, x \rangle + b_{r,0} - \tau| \leq k^{-1/2}]$ can be seen as $k$ independent Bernoulli random variables.

We have

$$\Pr[|\langle w_{r,0}, x \rangle + b_{r,0} - \tau| \leq k^{-1/2}] \leq \Pr[|\langle w_{r,0}, x \rangle + b_{r,0}| \leq k^{-1/2}]$$
$$\leq O(k^{-1/2}).$$

where the first step is due to Theorem A.3, the second step is due to Claim A.10.

The corresponding probability of $\mathbf{1}[|\langle w_{r,0}, x \rangle + b_{r,0} - \tau| \leq k^{-1/2}] = 1$ is bounded by $O(k^{-1/2})$. Therefore, by Chernoff bounds (Lemma A.8) with probability at least $1 - 1/\exp(\Omega(k^{1/2}))$, we have

$$\sum_{r=1}^{k} \mathbf{1}[|\langle w_{r,0}, x \rangle + b_{r,0} - \tau| \leq k^{-1/2}] \leq O(k^{1/2}). \tag{31}$$

With a union bound over two bounds in Eq. (30) and Eq. (31).

One can obtain the following bound for $S_1$:

$$S_1 \leq O(k^{-1/2} \cdot \mathcal{C}(q, \epsilon_3)).$$

The succeed probability of above equation is

$$1 - 1/\exp(\Omega(k^{1/2})) - 1/\exp(\Omega(k)) = 1 - 1/\exp(\Omega(k^{1/2})).$$

And now we will focus on getting the bound of $S_2$. We have

$$S_2 \leq \mathcal{C}(q, \epsilon_3) \underset{b \sim \mathcal{N}(0,1), w \sim \mathcal{N}(0, I_d)}{\mathbb{E}} [|\mathbf{1}[\langle w, x + \mu \rangle + b \geq \tau] - \mathbf{1}[\langle w, x \rangle + b \geq \tau]|]$$

$$\leq \mathcal{C}(q, \epsilon_3) \underset{b \sim \mathcal{N}(0,1), w \sim \mathcal{N}(0,I_d)}{\mathbb{E}} [\mathbf{1}[|\langle w, x \rangle + b| \leq k^{-1/2}] + \mathbf{1}[k^{1/2} \cdot c_2 < \|w\|_2]]$$

where the first step is because $|h(\cdot)| \leq \mathcal{C}(q, \epsilon_3)$, and the second step is because $|\mathbf{1}[\langle w, x + \mu \rangle + b \geq \tau] - \mathbf{1}[\langle w, x \rangle + b \geq \tau]|$ is decided by the sign of $\mathbf{1}[\langle w, x + \mu \rangle + b \geq \tau]$ and $\mathbf{1}[\langle w, x \rangle + b \geq \tau]$. The above difference is decided by the value of $\|\mu\|_2 \leq \frac{1}{k^c}$ and $w \sim \mathcal{N}(0, I_d)$, and $\mathbf{1}[k^{1/2} \cdot c_2 < \|w\|_2] \geq 0$. The constant $c_2$ is the same as previously.

Still,

$$\underset{b \sim \mathcal{N}(0,1), w \sim \mathcal{N}(0,I_d)}{\Pr}[k^{-1/2} \geq |\langle w, x \rangle + b - \tau|] \leq \underset{b \sim \mathcal{N}(0,1), w \sim \mathcal{N}(0,I_d)}{\Pr}[k^{-1/2} \geq |\langle w, x \rangle + b|]$$
$$\leq O(k^{-1/2})$$

where the first step is due to Theorem A.3, the second step is due to Lemma A.10.

Then we have

$$\underset{b \sim \mathcal{N}(0,1), w \sim \mathcal{N}(0,I_d)}{\Pr}[k^{1/2} \cdot c_2 < \|w\|_2] \leq 1/\exp(\Omega(k)).$$

Therefore,

$$S_2 \leq O(k^{-1/2} \cdot \mathcal{C}(q, \epsilon_3)).$$

This completes the proof. □

### C.16 PROOF OF TIME COMPLEXITY PER TRAINING ITERATION

We give the proof of Lemma 5.3 in this section.

*Proof.* The complexity for Algorithm 1 at each iteration can be decomposed as follows:

- Querying the active neuron set for $n$ adversarial training data points $\widetilde{x}_i$ takes $\widetilde{O}(m^{1-\Theta(1/d)}nd)$ time: $\sum_{i=1}^n \mathcal{T}_{\mathsf{query}}(2m, d, k_{i,t}) = \widetilde{O}(m^{1-\Theta(1/d)}nd)$ according to Corollary 5.2.

- Forward computation takes $O(d \cdot (Q_0 + nm^{7/8}))$ time: $\sum_{i=1}^n O(d \cdot k_{i,t}) = O(d \cdot (Q_0 + nm^{7/8}))$ according to Lemma 4.5 and Claim 4.6.

- Backward computation involving computing gradient $\Delta W$ and updating $W_{t+1}$ takes $O(d \cdot (Q_0 + nm^{7/8}))$ time: $O(d \cdot \mathrm{nnz}(P)) = O(d \cdot (Q_0 + nm^{7/8}))$ according to Lemma 4.5 and Claim 4.6.

- Updating the weight vectors takes $O((Q_0 + nm^{7/8}) \cdot \log^2(2m))$ time.
$$\mathcal{T}_{\mathsf{update}} \cdot (|Q_{t,i}| + |Q_{t+1,i}|)$$
$$= O(\log^2(2m)) \cdot (\sum_{i=1}^n k_{i,t} + k_{i,t+1})$$
$$= O(\log^2(2m)) \cdot (Q_0 + O(nm^{7/8}))$$
$$= O((Q_0 + nm^{7/8}) \cdot \log^2(2m))$$
where the first step is according to Corollary 5.2, the second step is the result of Lemma 4.5 and Claim 4.6, and the third step is the result of aggregating the terms.

Summing over all the quantities in four bullets gives us the per iteration cost $\widetilde{O}(m^{1-\Theta(1/d)}nd)$. □

