# OpenReview forum: "A Sublinear Adversarial Training Algorithm"
_ICLR.cc/2024/Conference — ICLR 2024 poster_

### Official Review · Reviewer_5dfS · 2023-10-31

**Soundness:** 2 fair
**Presentation:** 1 poor
**Contribution:** 2 fair
**Rating:** 3
**Confidence:** 3

**Summary:**

The authors consider fully connected one hidden layer neural networks with shifted ReLU activations and develop an adversarial training algorithm with cost $o(mnd)$ per iteration by applying half-space reporting data structure. To do that, the authors rely on three main techniques:
1. approximation via pseudo-network sufficiently precise with high probability
2. using shifted ReLU functions instead of standard ones to attain $o(mnd)$ time complexity per training iteration
3. use a tree data structure to find the neurons that need updating more quickly (Half-space reporting)

The algorithm consists of a loop that first calculates the adversarial training set and the indices of the activated neurons for each element of the training set, then doing a gradient update for all active neurons. A series of theorems and lemmata are presented, analyzing the proposed algorithm, which is followed by a time complexity analysis. The reviewer did not check the proofs in the appendix.

**Strengths:**

- Adversarial training algorithms and their analysis is an important problem in the field of adversarial robustness.
- The assumptions seem reasonable.

**Weaknesses:**

- The relevance of the paper remains unclear. Is it theoretically relevant or also practically?
- The restriction to 1 hidden layer networks is strong.
- The Paper is hard to read, the reasoning jumps back and forth. The reviewer recommends a thorough rewrite. It would be good if the authors first introduce concepts and then use them. An example is the "query, insert, remove" from the data structure in Algorithm 1 (Page 6) that gets introduced in Section 5 (Page 8). Similarly, Theorem 1.1 (Page 2) has a reference to Equation 2 (Page 4). The paper seems to be rushed, also reads that way.

Typos:
	- 3.1 the typesetting of $\ell_p$
	- The inequality in Theorem 4.1 seems to be the wrong way round.
	- In theorem 4.3: "$T = \Theta(\epsilon^{-2} K^2)$. in Algorithm"

**Questions:**

- Are the contributions of the paper mainly of theoretical or also practical interest? Can the algorithm be efficiently parallelized? Are efficient vectorized/GPU implementations possible? What are the memory requirements of the algorithm?
- Page 3: what is the definition for an $\epsilon$-net?
- What about deeper networks? On an intuitive level, the reviewer does not see a reason that the complexity should worsen there.

---

> ### Author Response · Authors · 2023-11-20
>
> Thank you for your meticulous review and valuable suggestions. We will revise the typos in the next version. We would like to respond to your questions respectively:
> 1. Our paper is theoretically relevant and analyzes the convergence guarantee of adversarial training procedures on a two-layer neural network with shifted ReLU activation, demonstrating that only a sublinear number of neurons (o(m)) are activated for each input data per iteration​ and developing an algorithm for adversarial training with time cost o(mnd) per iteration by applying half-space reporting data structure.
> 2. The generalization of our method to multiple layers is a very interesting question. However, as the main purpose of our submission is to build an algorithm with guaranteed performance and rigorous analysis, we focus on over-parametrized two-layer neural network, which is a common object studied in many recent works about neural tangent kernel [1,2,3,4,5,6] and theoretical papers about deep learning [7,8,9]. We also note that there are new empirical results on multi-layer neural network settings [10,11].  We highly appreciate such efforts and would make theoretical analysis of multiple-layer neural network settings our future work.
> 3. We thank you for pointing out the writing improvement suggestions. We will update the overall structure in the next version.
>
> As for your questions:
> 1. The contribution of this paper is a theoretical. The introduced half-space reporting data structure used in this paper will take additional $O(n \log n)$ space [12] . To have an intuitive understanding of the level of such additional memory usage, we would point out that the weights of neural networks takes $O(n d)$ space. So when the neural network is over-parameterized, i.e., $d$ is very large, the additional memory usage by our algorithm is not the dominant term in space complexity.
> 2. The definition of $\epsilon$-net is defined as follows: Given a metric space $(X, d)$, a subset $Y$ of $X$ is said to be an  $\epsilon$-net if
>
>     (i) For $a, b \in Y$ , we have $d(a, b) \ge \epsilon$, and
>
>     (ii) For all $x \in X$, there exists an $a \in Y$ such that $d(x, a) < \epsilon$.
> 3. As for deeper NN, please refer to point (2) in the weakness part response.
>
> [1] Zeyuan Allen-Zhu, Yuanzhi Li, and Yingyu Liang. Learning and generalization in overparameterized neural networks, going beyond two layers. In NeurIPS. arXiv preprint arXiv:1811.04918, 2019a.
>
> [2] Arthur Jacot, Franck Gabriel, and Cl  ́ement Hongler. Neural tangent kernel: Convergence and generalization in neural networks. In Advances in neural information processing systems, pages 8571–8580, 2018.
>
> [3] Zeyuan Allen-Zhu, Yuanzhi Li, and Zhao Song. A convergence theory for deep learning via over-parameterization. In ICML, 2019b.
>
> [4] Zeyuan Allen-Zhu, Yuanzhi Li, and Zhao Song. On the convergence rate of training recurrent neural networks. In NeurIPS, 2019c
>
> [5] Yi Zhang, Orestis Plevrakis, Simon S Du, Xingguo Li, Zhao Song, and Sanjeev Arora. Over-parameterized adversarial training: An analysis overcoming the curse of dimensionality. Advances in Neural Information Processing Systems, 33:679–688, 2020.
>
> [6] Simon S Du, Xiyu Zhai, Barnabas Poczos, and Aarti Singh. Gradient descent provably optimizes over-parameterized neural networks. In ICLR, 2019.
>
> [7] Zhong, Kai, et al. "Recovery guarantees for one-hidden-layer neural networks." International conference on machine learning. PMLR, 2017.
>
> [8] Du, Simon, et al. "Gradient descent finds global minima of deep neural networks." International conference on machine learning. PMLR, 2019.
>
> [9] Mei, Song, Andrea Montanari, and Phan-Minh Nguyen. "A mean field view of the landscape of two-layer neural networks." Proceedings of the National Academy of Sciences 115.33 (2018): E7665-E7671.
>
> [10] Beidi Chen, Tharun Medini, James Farwell, Charlie Tai, Anshumali Shrivastava, et al. Slide: In defense of smart algorithms over hardware acceleration for large-scale deep learning systems. Proceedings of Machine Learning and Systems, 2:291–306, 2020
>
> [11] Beidi Chen, Zichang Liu, Binghui Peng, Zhaozhuo Xu, Jonathan Lingjie Li, Tri Dao, Zhao Song, Anshumali Shrivastava, and Christopher Re. Mongoose: A learnable lsh framework for efficient neural network training. In International Conference on Learning Representations, 2021.
>
> [12] Pankaj K Agarwal, David Eppstein, and Jiri Matousek. Dynamic half-space reporting, geometric optimization, and minimum spanning trees. In Annual Symposium on Foundations of Computer Science, volume 33, pages 80–80. IEEE COMPUTER SOCIETY PRESS, 1992

---

### Official Review · Reviewer_RwpB · 2023-11-01

**Soundness:** 3 good
**Presentation:** 3 good
**Contribution:** 2 fair
**Rating:** 6
**Confidence:** 3

**Summary:**

The paper proposes an adversarial training algorithm for 2-layer neural networks with ReLU activation that has sublinear o(mnd) per-iteration time complexity, compared to standard Ω(mnd). The result mainly relies on the insight that only o(m) neurons will be activated for each input data per iteration.

**Strengths:**

1. The paper presents some interesting result on analyzing per-iteration complexity of adversarial training of a special setting of 2-layer neural networks. Theoretical analyses are solid and insightful.
2. Complexity of adversarial training is less analyzed compared with standard training, the work would be interesting to some researchers.

**Weaknesses:**

1. Just like in many other works on 2 layer neural nets, it is unclear if the insight can generalize to the more common setting of deeper neural nets.
2. It seems only optimal adversaries are analyzed (e.g., in theorem 4.4), but most cases it is impractical to have an optimal adversary.

**Questions:**

Could authors provide some discussions on and thoughts on if the activation sparsity could generalize to deeper nets?

---

> ### Author Response · Authors · 2023-11-20
>
> We deeply appreciate your insightful feedback and are grateful for the time you took to share it.
>
> 1. This is a very interesting future direction. We also note that there are new empirical works on multi-layer neural network settings [1, 2] that find a similar phenomenon of sparsity that inspired this work.  We highly appreciate such efforts and would make theoretical analysis of multiple layer neural network setting as our future work.
> 2. We would point out that if the adversarial example construction is not optimal, it does not hurt the acceleration achieved by our algorithm. However, we also admit that if we are not using the optimal adversarial construction, then the error bound in Theorem 4.4 for weights will be worse. Depending on what specific assumptions we make for the non-optimal adversarial examples, e.g. we are using adversarial construction $A$ instead of the optimal one $A^*$, the error bound for $W$ will be correspondingly worse.
> 3. As mentioned in point 1, some empirical works observe a similar phenomenon of sparsity in multi-layer neural networks. Hence the intuition might still be avoiding making redundant $0$-update for non-activated neurons. The major difficulty would then be designing new data structures that could efficiently maintain activated neurons during the adversarial training procedure, which we make as a very interesting future direction.
>
>
> [1] Beidi Chen, Tharun Medini, James Farwell, Charlie Tai, Anshumali Shrivastava, et al. Slide: In defense of smart algorithms over hardware acceleration for large-scale deep learning systems. Proceedings of Machine Learning and Systems, 2:291–306, 2020
>
> [2] Beidi Chen, Zichang Liu, Binghui Peng, Zhaozhuo Xu, Jonathan Lingjie Li, Tri Dao, Zhao Song, Anshumali Shrivastava, and Christopher Re. Mongoose: A learnable lsh framework for efficient neural network training. In International Conference on Learning Representations, 2021.

---

### Official Review · Reviewer_Wuuy · 2023-11-01

**Soundness:** 4 excellent
**Presentation:** 3 good
**Contribution:** 3 good
**Rating:** 6
**Confidence:** 3

**Summary:**

The authors propose an adversarial training algorithm designed for two-layer ReLU networks. Combining multiple techniques, they manage to prove that their algorithm runs in time sublinear in the size of the network.

**Strengths:**

- The authors propose an interesting algorithm and manage to show that per iteration cost is sublinear in the size of the network for a two-layer ReLU network. The paper is very clearly written with detailed proofs in the appendix.

- The use of the techniques such as half-space reporting data structures and shifted ReLU in the design of the algorithm are interesting and instructive.

**Weaknesses:**

- The current result seems to be more of a theoretical interest than a practical algorithm. The limitation of two-layer network, well-separation for adversarial loss are both severe limitations in practice.

- There are also some unclear or missing details that need to be addressed in the Questions section below.

**Questions:**

- Is there any assumption on the adversarial example oracle A in Algorithm 1? I don't seem be able to find any reference to its complexity analysis in the main paper. Usually adversarial examples are found by iterative projected gradient descent (on the input x) and their runtime should also be taken into account.

- If we take away the adversarial example generation in Algorithm 1 and just use the input x_i as training example, we should then obtain a sublinear training algorithm for a two-layer ReLU network. I am wondering why the authors are focusing on adversarial training, or if there are any parts of the algorithm that are specific to the problem of adversarial examples. The 3 techniques listed in Section 2 are not specific to adversarial training.

---

> ### Author Response · Authors · 2023-11-20
>
> Thank you for your valuable questions. We would revise the typos in the next version.
> 1. We take standard adversarial example construction in our algorithm. So if we let $T$ denote the complexity for the adversarial example oracle, then the improvement our algorithm achieved is $o(nmdT)$ compared to standard algorithm $\Omega(nmdT)$.
> 2. As mentioned above, if we let $T=O(1)$, i.e., use $x_i$ as training example, then our algorithm achieves $o(nmd)$ compared to the well-known standard result $\Omega(nmd)$.

---

### Official Review · Reviewer_yRgF · 2023-11-05

**Soundness:** 3 good
**Presentation:** 3 good
**Contribution:** 3 good
**Rating:** 8
**Confidence:** 3

**Summary:**

The paper tries to address the computational complexity of adversarial training in neural networks, which is an active research area.
The work addresses training efficiency by trading off extra memory for computation and incorporating an additional half-space data structure, efficiently identifying only the active neurons for computation. The key insight is that not all neurons are active for every input data point, enabling the computation of the dot product exclusively with the active neurons. While the method of initially identifying active neurons is not novel, the application of efficient geometric data structures is both new and promising. The research introduces an algorithm designed for this purpose; although it does not include experimental results, it does offer convergence proofs for two-layer networks.

**Strengths:**

The approach is novel where adversarial training that leverages geometric data structures at the expense of extra space. It extends the direction of first identifying the active neurons and then only taking dot product with those neurons.

The paper is well written and however, without empirical results or illustrative examples, the clarity of the paper's practical implications could be enhanced.

The work is significant as it has a potential to further improve the direction of efficient adversarial training of robust neural networks.

I have not conducted an exhaustive review of the convergence proofs; however, upon initial examination, they appear to be correct.

**Weaknesses:**

The study lacks experiments, even on toy datasets, to determine whether its findings are applicable in practical scenarios.

Furthermore, it has not been compared with existing works. Although it is primarily theoretical, clarity on how it measures up against established methods is missing.

The paper does not discuss how reducing training time might impact the robustness performance of the model. This trade-off between efficiency and robustness is crucial for practical use cases.

Likewise, the trade-off between memory usage and efficiency warrants discussion. While the paper could comment on memory requirements, it may be that for only two-layer networks, the difference is minimal. However, it would still be beneficial to understand this aspect, perhaps by employing very wide neural networks to confirm it in a practical scenario.

It is also unclear whether the proposed methods can be effectively translated into actual GPU computations. If the approach results in underutilized GPUs, it might still be valuable but may not reduce the overall wall time. Some commentary on this feasibility would be constructive.

**Questions:**

To translate the theoretical advantages into practical scenarios, the paper should present at least some toy experiments.

It is important to explore how reducing training time affects the model's robustness performance, as the trade-off between efficiency and robustness is crucial for practical use cases.

The same consideration should be given to the trade-off between memory usage and efficiency. It would be beneficial for the paper to comment on memory requirements. While the impact on only a two-layer network might be minor, understanding this trade-off is essential, and could be tested using neural networks with extremely high widths in practical scenarios.

In short, a toy experiment featuring a two-layer network that showcases practical, minimal use cases by comparing memory usage, computational efficiency, and robustness performance would be a valuable addition to the work.

---

> ### Author Response · Authors · 2023-11-20
>
> Thank you for your valuable feedback. We would revise the typos in the next version. We admit it is very important to take memory usage into consideration in this work. The introduced half-space reporting data structure used in this paper will take additional $O(n \log n)$ space [1] . To have an intuitive understanding of the level of such additional memory usage, we would point out that the weights of neural networks takes $O(n d)$ space. So when the neural network is over-parameterized, i.e., $d$ is very large, the additional memory usage by our algorithm is not the dominant term in space complexity.
>
> [1] Pankaj K Agarwal, David Eppstein, and Jiri Matousek. Dynamic half-space reporting, geometric optimization, and minimum spanning trees. In Annual Symposium on Foundations of Computer Science, volume 33, pages 80–80. IEEE COMPUTER SOCIETY PRESS, 1992

---

### Meta-Review · Area_Chair_1QoB · 2023-12-15

**Metareview:**

This paper analyzes the convergence guarantee of the adversarial training procedure on a two-layer neural network. Three reviewers have found the work interesting while another reviewer has been highly critical of this paper. The main issue is the restriction to two-layer neural networks. It is not clear if the results could be extended to a multi-layer NN and, if not, how useful the developed results are. There are also some concerns about the presentation of the paper.

**Justification For Why Not Higher Score:**

I reached this decision by evaluating the contributions and novelty of the work, taking into consideration both the reviews and the responses from the authors.

**Justification For Why Not Lower Score:**

I reached this decision by evaluating the contributions and novelty of the work, taking into consideration both the reviews and the responses from the authors.

---

### Decision · Program_Chairs · 2024-01-16

Accept (poster)